# Efficient Adversarial Contrastive Learning via Robustness-Aware Coreset Selection

**Xilie Xu**[1*]**, Jingfeng Zhang**[2,3*†]**, Feng Liu**[4]**, Masashi Sugiyama**[2,5]**, Mohan Kankanhalli**[1]

[1] School of Computing, National University of Singapore
[2] RIKEN Center for Advanced Intelligence Project (AIP)
[3] School of Computer Science, The University of Auckland
[4] School of Computing and Information Systems, The University of Melbourne
[5] Graduate School of Frontier Sciences, The University of Tokyo
xuxilie@comp.nus.edu.sg    jingfeng.zhang@auckland.ac.nz
fengliu.ml@gmail.com    sugi@k.u-tokyo.ac.jp
mohan@comp.nus.edu.sg

## Abstract

Adversarial contrastive learning (ACL) does not require expensive data annotations but outputs a robust representation that withstands adversarial attacks and also generalizes to a wide range of downstream tasks. However, ACL needs tremendous running time to generate the adversarial variants of all training data, which limits its scalability to large datasets. To speed up ACL, this paper proposes a *robustness-aware coreset selection* (RCS) method. RCS does not require label information and searches for an informative subset that minimizes a representational divergence, which is the distance of the representation between natural data and their virtual adversarial variants. The vanilla solution of RCS via traversing all possible subsets is computationally prohibitive. Therefore, we theoretically transform RCS into a surrogate problem of submodular maximization, of which the greedy search is an efficient solution with an optimality guarantee for the original problem. Empirically, our comprehensive results corroborate that RCS can speed up ACL by a large margin without significantly hurting the *robustness transferability*. Notably, to the best of our knowledge, we are the first to conduct ACL efficiently on the large-scale ImageNet-1K dataset to obtain an effective robust representation via RCS. Our source code is at https://github.com/GodXuxilie/Efficient_ACL_via_RCS.

## 1 Introduction

The pre-trained models can be easily finetuned to downstream applications, recently attracting increasing attention [1, 2, 3, 4]. Notably, vision Transformer [5] pre-trained on ImageNet-1K [1] can achieve state-of-the-art performance on many downstream computer-vision applications [6, 7]. Foundation models [8] trained on large-scale unlabeled data (such as GPT [9] and CLAP [10]) can be adapted to a wide range of downstream tasks. Due to the prohibitively high cost of annotating large-scale data, the pre-trained models are commonly powered by the techniques of unsupervised learning [11, 12] in which *contrastive learning* (CL) is the most effective learning style to obtain the generalizable feature representations [13, 14].

*Adversarial CL* (ACL) [14, 15, 16, 17], that incorporate adversarial data with contrastive loss, can yield a robust representation that is adversarially robust. Compared with the standard CL [13], ACL can output a robust representation that can transfer some robustness to the downstream tasks against

---

[*]The first two authors have made equal contributions.
[†]Corresponding author.

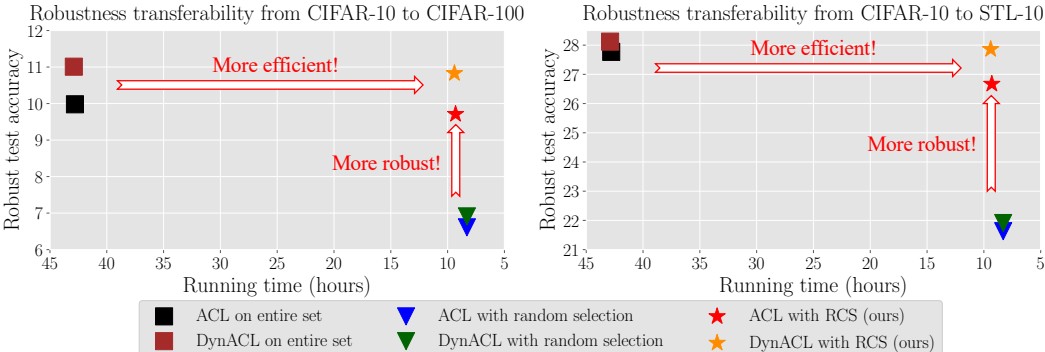

Figure 1: We learn a representation using CIFAR-10 [6] dataset (without requiring labels) via ACL [14] and DynACL [17]. Then, we evaluate the representation's robustness transferability to CIFAR-100 [6] and STL-10 [26] (using labels during finetuning) via standard linear finetuning. We demonstrate the running time of robust pre-training w.r.t. different coreset selection (CS) strategies and report the robust test accuracy under AutoAttack [19]. Our proposed RCS (star shapes) always achieves a higher robust test accuracy than random selection (triangles) while consuming much less running time than pre-training on the entire set (squares). Experimental details are in Appendix B.4.

adversarial attacks [18, 19] via finetuning. The robustness transferability is of great importance to a pre-trained model's practicality in safety-critical downstream tasks [20, 21, 22, 23].

However, ACL is computationally expensive, which limits its scalability. At each training epoch, ACL first needs to conduct several backward propagations (BPs) on all training data to generate their adversarial variants and then train the model with those adversarial data. Even if we use the techniques of fast adversarial training [24, 25] that significantly reduce the need for conducting BPs per data, ACL still encounters the issue of the large scale of the training set that is commonly used for pre-training a useful representation.

*Coreset selection* (CS) [27, 28] that selects a small yet informative subset can reduce the need for the whole training set, but CS cannot be applied to ACL directly. Mirzasoleiman et al. [29] and Killamsetty et al. [30, 31, 32] have significantly accelerated the standard training by selecting an informative training subset. Recently, Dolatabadi et al. [33] proposed an adversarial CS method to accelerate the standard adversarial training [34, 35, 24] by selecting a representative subset that can accurately align the gradient of the full set. However, the existing CS methods require the label information of training data, which are not applicable to the ACL that learns from the unlabeled data.

To accelerate the ACL, this paper proposes a robustness-aware coreset selection (RCS) that selects a coreset without requiring the label information but still helps ACL to obtain an effective robust representation. RCS searches for a coreset that minimizes the representational divergence (RD) of a representation. RD measures the representation difference between the natural data and their virtual adversarial counterparts [36], in which virtual adversarial data can greatly alter the output distribution in the sense of feature representation.

Although solving the RCS via traversing all possible subsets is simple, it is computationally prohibitive. Therefore, we transform the RCS into a proxy problem of maximizing a set function that is theoretically shown monotone and $\gamma$-weakly submodular (see Theorem 1) subject to cardinality constraints [37, 38]. Then, we can apply the greedy search [37] to efficiently find a coreset that can minimize the RD. Notably, Theorem 2 provides the theoretical guarantee of optimality of our greedy-search solution.

Empirically, we demonstrate RCS can indeed speed up ACL [14] and its variants [17, 15]. As shown in Figure 1, RCS can speed up both ACL [14] and DynACL [17] by 4 times and obtain an effective representation without significantly hurting the *robustness transferability* (details in Section 4.1). Notably, to the best of our knowledge, we are the first to apply ACL on the large-scale ImageNet-1K [1] dataset to obtain a robust representation via RCS (see Section 4.2). Besides, we empirically show that RCS is compatible with standard adversarial training [34, 35, 24] as well in Section 4.3. Our comprehensive experiments corroborate that our proposed RCS is a unified and principled framework for efficient robust learning.

## 2    Background and Preliminaries

In this section, we introduce related work and preliminaries about contrastive learning [13, 14, 17].

### 2.1    Related Work

**Contrastive learning (CL).**    CL approaches are frequently used to leverage large unlabeled datasets for learning useful representations. Previous unsupervised methods that map similar samples to similar representations [39] have the issue of collapsing to a constant representation. CL addresses the representational collapse by introducing the negative samples [12]. Chen et al. [13] presented SimCLR that leverages the contrastive loss for learning useful representations and achieved significantly improved accuracy on the standard suit of downstream tasks. Recently, adversarial contrastive learning (ACL) [40, 14, 41, 15, 16, 42, 17, 43] that incorporates adversarial training with the contrastive loss [13] has become the most effective unsupervised approaches to learn robust representations. Jiang et al. [14] showed that ACL could exhibit better adversarial robustness on downstream tasks compared with standard CL (i.e., SimCLR [13]). Luo et al. [17] proposed to dynamically schedule the strength of data augmentations to narrow the gap between the distribution of training data and that of test data, thus enhancing the performance of ACL. Xu et al. [43] improved ACL from the lens of causality [44, 45] and proposed an adversarial invariant regularization that achieves new SOTA results.

However, ACL needs large computational resources to obtain useful representations from large-scale datasets. ACL needs to spend a large amount of running time generating adversarial training data during pre-training. Fast adversarial training (Fast-AT) [24] uses the fast gradient descent method (FGSM) [18] to accelerate the generation procedure of adversarial training data, thus speeding up robust training. A series of recent works [46, 47] followed this line and further improved the performance of fast AT. However, the time complexity of ACL is still proportional to the size of the training set. Besides, the large-scale training sets (e.g., ImageNet-1K [1] contains over 1 million training images) further contribute to the inefficiency of adversarially pre-training procedures [48, 49]. To enable efficient ACL, we propose a novel CS method that can speed up ACL by decreasing the amount of training data.

**Coreset selection (CS).**    CS aims to select a small yet informative data subset that can approximate certain desirable characteristics (e.g., the loss gradient) of the entire set [28]. Several studies have shown that CS is effective for efficient standard training in supervised [29, 31, 30] and semi-supervised [32] settings. Dolatabadi et al. [33] proposed adversarial coreset selection (ACS) that selects representative coresets of adversarial training data that can estimate the adversarial loss gradient on the entire training set. However, ACS is only adapted to supervised AT [34, 35]. Thus, ACS requires the label information and is not applicable to ACL. In addition, previous studies did not explore the influence of CS on the pre-trained model's transferability. To this end, we propose a novel CS method that does not require label information and empirically demonstrate that our proposed method can significantly accelerate ACL while only slightly hurting the transferability.

### 2.2    Preliminaries

Here, we introduce the preliminaries of contrastive learning [13, 14, 17]. Let $(\mathcal{X}, d_\infty)$ be the input space $\mathcal{X}$ with the infinity distance metric $d_\infty(x, x') = \|x - x'\|_\infty$, and $\mathcal{B}_\epsilon[x] = \{x' \in \mathcal{X} \mid d_\infty(x, x') \leq \epsilon\}$ be the closed ball of radius $\epsilon > 0$ centered at $x \in \mathcal{X}$.

**ACL [14] and DynACL [17].**    We first introduce the standard contrastive loss [13]. Let $f_\theta : \mathcal{X} \to \mathcal{Z}$ be a feature extractor parameterized by $\theta$, $g : \mathcal{Z} \to \mathcal{V}$ be a projection head that maps representations to the space where the contrastive loss is applied, and $\tau_i, \tau_j : \mathcal{X} \to \mathcal{X}$ be two transformation operations randomly sampled from a pre-defined transformation set $\mathcal{T}$. Given a minibatch $B \sim \mathcal{X}^\beta$ consisting of $\beta$ samples, we denote the augmented minibatch $B' = \{\tau_i(x_k), \tau_j(x_k) \mid \forall x_k \in B\}$ consisting of $2\beta$ samples. We take $h_\theta(\cdot) = g \circ f_\theta(\cdot)$ and $x_k^u = \tau_u(x_k)$ for any $x_k \sim \mathcal{X}$ and $u \in \{i, j\}$. The standard contrastive loss of a positive pair $(x_k^i, x_k^j)$ is as follows:

$$\ell_{\mathrm{CL}}(x_k^i, x_k^j; \theta) = - \sum_{u \in \{i,j\}} \log \frac{e^{\mathrm{sim}\left(h_\theta(x_k^i), h_\theta(x_k^j)\right)/t}}{\sum\limits_{x \in B' \backslash \{x_k^u\}} e^{\mathrm{sim}\left(h_\theta(x_k^u), h_\theta(x)\right)/t}},$$

where $\mathrm{sim}(p, q) = p^\top q / \|p\|\|q\|$ is the cosine similarity function and $t > 0$ is a temperature parameter.

Given an unlabeled set $X \sim \mathcal{X}^N$ consisting of $N$ samples, the loss function of ACL [14] is $\mathcal{L}_{\mathrm{ACL}}(X;\theta) = \sum_{k=1}^{N} \ell_{\mathrm{ACL}}(x_k;\theta)$ where

$$\ell_{\mathrm{ACL}}(x_k;\theta) = \Big( \max_{\substack{\tilde{x}_k^i \in \mathcal{B}_\epsilon[x_k^i] \\ \tilde{x}_k^j \in \mathcal{B}_\epsilon[x_k^j]}} (1+\omega) \cdot \ell_{\mathrm{CL}}(\tilde{x}_k^i, \tilde{x}_k^j; \theta) \Big) + (1-\omega) \cdot \ell_{\mathrm{CL}}(x_k^i, x_k^j; \theta),$$

in which $\omega \in [0,1]$ is a hyperparameter and $\tilde{x}_k^i$ and $\tilde{x}_k^j$ are adversarial data generated via projected gradient descent (PGD) [34] within the $\epsilon$-balls centered at $x_k^i$ and $x_k^j$. Note that ACL [14] fixes $\omega = 0$ while DynACL [17] dynamically schedules $\omega$ according to its dynamic augmentation scheduler that gradually anneals from a strong augmentation to a weak one. We leave the details of the data augmentation scheduler [17] in Appendix B.2 due to the limited space.

Given an initial positive pair $(x^{i,(0)}, x^{j,(0)})$, PGD step $T \in \mathbb{N}$, step size $\rho > 0$, and adversarial budget $\epsilon \geq 0$, PGD iteratively updates the pair of data from $\tau = 0$ to $T-1$ as follows:

$$x^{i,(\tau+1)} = \Pi_{\mathcal{B}_\epsilon[x^{i,(0)}]}\big(x^{i,(\tau)} + \rho \, \mathrm{sign}(\nabla_{x^{i,(\tau)}} \ell_{\mathrm{CL}}(x^{i,(\tau)}, x^{j,(\tau)}))\big),$$
$$x^{j,(\tau+1)} = \Pi_{\mathcal{B}_\epsilon[x^{j,(0)}]}\big(x^{j,(\tau)} + \rho \, \mathrm{sign}(\nabla_{x^{j,(\tau)}} \ell_{\mathrm{CL}}(x^{i,(\tau)}, x^{j,(\tau)}))\big),$$

where $\Pi_{\mathcal{B}_\epsilon[x]}$ projects the data into the $\epsilon$-ball around the initial point $x$.

ACL is realized by conducting one step of inner maximization on generating adversarial data via PGD and one step of outer minimization on updating $\theta$ by minimizing the ACL loss on generated adversarial data, alternatively. Note that the parameters of the projection head $g$ are updated as well during ACL. Here we omit the parameters of $g$ for notational simplicity since we only use the parameters of feature extractor $f_\theta$ on downstream tasks after completing ACL.

## 3 Robustness-Aware Coreset Selection

In this section, we first introduce the representational divergence (RD), which is quantified by the distance of the representation between natural data and their virtual adversarial variants, as the measurement of the adversarial robustness of a representation. Then, we formulate the learning objective of the Robustness-aware Coreset Selection (RCS). Next, we theoretically show that our proposed RCS can be efficiently solved via greedy search. Finally, we give the algorithm of efficient ACL via RCS.

### 3.1 Representational Divergence (RD)

Adversarial robustness is the most significant property of adversarially pre-trained models, which could lead to enhanced robustness transferability. Inspired by previous studies [35, 50], we measure the adversarial robustness of the representation in an unsupervised manner using the representational divergence (RD). Given a natural data point $x \sim \mathcal{X}$ and a model $g \circ f_\theta : \mathcal{X} \to \mathcal{V}$ composed of a feature extractor $f_\theta$ and a projector head $g$, RD of this data point $\ell_{\mathrm{RD}}(x;\theta)$ is quantified by the distance between the representation of the natural data and that of its virtual adversarial counterpart [36], i.e.,

$$\ell_{\mathrm{RD}}(x;\theta) = d(g \circ f_\theta(\tilde{x}), g \circ f_\theta(x)) \quad \text{s.t.} \quad \tilde{x} = \arg\max_{x' \in \mathcal{B}_\epsilon[x]} d(g \circ f_\theta(x'), g \circ f_\theta(x)), \qquad (1)$$

in which we can use PGD method [51] to generate adversarial data $\tilde{x}$ within the $\epsilon$-ball centered at $x$ and $d(\cdot, \cdot) : \mathcal{V} \times \mathcal{V} \to \mathbb{R}$ is a distance function, such as the Kullback–Leibler (KL) divergence [35], the Jensen-Shannon (JS) divergence [52], and the optimal transport (OT) distance [50]. We denote the RD on the unlabeled validation set $U$ as $\mathcal{L}_{\mathrm{RD}}(U;\theta) = \sum_{x_i \in U} \ell_{\mathrm{RD}}(x_i;\theta)$. The smaller the RD is, the representations are of less sensitivity to adversarial perturbations, thus being more robust.

### 3.2 Learning Objective of Robustness-Aware Coreset Selection (RCS)

Our proposed RCS aims to select an informative subset that can achieve the minimized RD between natural data and their adversarial counterparts, thus expecting the selected coreset to be helpful in improving the adversarial robustness of representations. Therefore, given an unlabeled training set $X \sim \mathcal{X}^N$ and an unlabeled validation set $U \sim \mathcal{X}^M$ ($M \ll N$), our proposed Robustness-aware Coreset Selection (RCS) searches for a coreset $S^*$ such that

$$S^* = \arg\min_{S \subseteq X, |S|/|X| \leq k} \mathcal{L}_{\mathrm{RD}}(U; \arg\min_\theta \mathcal{L}_{\mathrm{ACL}}(S;\theta)). \qquad (2)$$

Note that the subset fraction $k \in (0, 1]$ controls the size of the coreset, i.e., $|S^*| \leq kN$. RCS only needs to calculate RD on the validation set (i.e., $\mathcal{L}_{\mathrm{RD}}(U)$) and ACL loss on the subset (i.e., $\mathcal{L}_{\mathrm{ACL}}(S)$), which can be computed in an unsupervised manner. Thus, RCS is applicable to ACL on unlabeled datasets, and compatible with supervised AT on labeled datasets as well, such as Fast-AT [24], SAT [34] and TRADES [35] (details in Appendix B.13). Intuitively, the coreset $S^*$ found by RCS can make the pre-trained model via minimizing the ACL loss $\mathcal{L}_{\mathrm{ACL}}(S^*)$ achieve the minimized $\mathcal{L}_{\mathrm{RD}}(U)$, thus helping attain adversarially robust presentations, which could be beneficial to the robustness transferability to downstream tasks.

To efficiently solve the inner minimization problem of Eq. (2), we modify Eq. (2) by conducting a one-step gradient approximation as the first step, which keeps the same practice as [31, 30], i.e.,

$$S^* = \underset{S \subseteq X, |S|/|X| \leq k}{\arg\min} \mathcal{L}_{\mathrm{RD}}(U; \theta - \eta \nabla_\theta \mathcal{L}_{\mathrm{ACL}}(S; \theta)), \tag{3}$$

where $\eta > 0$ is the learning rate. We define a set function $G : 2^X \to \mathbb{R}$ as follows:

$$G_\theta(S \subseteq X) \triangleq -\mathcal{L}_{\mathrm{RD}}(U; \theta - \eta \nabla_\theta \mathcal{L}_{\mathrm{ACL}}(S; \theta)). \tag{4}$$

We fix the size of the coreset, i.e., $|S^*| = k|X|$ being a constant. Then, the optimization problem in Eq. (2) can be reformulated as

$$S^* = \underset{S \subseteq X, |S|/|X| = k}{\arg\max} G_\theta(S). \tag{5}$$

### 3.3 Method—Greedy Search

Solving Eq. (5) is formulated as maximizing a set function subject to a cardinality $|S|$ constraint [53, 31, 32]. A naive solution to this problem is to traverse all possible subsets of size $kN$ and select the subset $S$ which has the largest value $G_\theta(S)$. But this naive solution is computationally prohibitive since it needs to calculate the loss gradient $\nabla_\theta \mathcal{L}_{\mathrm{ACL}}(S; \theta)$ for $\binom{N}{kN}$ times. Fortunately, Das and Kempe [37] and Gatmiry and Gomez-Rodriguez [38] proposed the greedy search algorithm to efficiently solve this kind of problem approximately if the set function is monotone and $\gamma$-weakly submodular. Note that the greedy search algorithm shown in Algorithm 1 only needs to calculate the loss gradient $\lceil N/\beta \rceil + \lfloor kN/\beta \rfloor \ll \binom{N}{kN}$ times where $\beta$ is batch size.

**Definition 1** (Monotonicity and $\gamma$-weakly submodularity [37, 54]). *Given a set function $G : 2^X \to \mathbb{R}$, the marginal gain of $G$ is defined as $G(x|A) \triangleq G(A \cup \{x\}) - G(A)$ for any $A \subset X$ and $x \in X \setminus A$. The set function $G$ is monotone if $G(x|A) \geq 0$ for any $A \subset X$ and $x \in X \setminus A$. The set function $G$ is called $\gamma$-weakly submodular if $\sum_{x \in B} G(x|A) \geq \gamma[G(A \cup B) - G(A)]$ for some $\gamma \in [0, 1]$ and any disjoint subset $A, B \subseteq X$.*

**Assumption 1.** *The first-order gradients and the second-order gradients of $\ell_{\mathrm{RD}}$ and $\ell_{\mathrm{ACL}}$ are bounded w.r.t. $\theta$, i.e.,*

$$\|\frac{\partial}{\partial_\theta} \ell_{\mathrm{RD}}(x; \theta)\| \leq L_1, \|\frac{\partial}{\partial_\theta} \ell_{\mathrm{ACL}}(x; \theta)\| \leq L_2, \|\frac{\partial^2}{\partial_\theta} \ell_{\mathrm{RD}}(x; \theta)\| \leq L_3, \|\frac{\partial^2}{\partial_\theta} \ell_{\mathrm{ACL}}(x; \theta)\| \leq L_4,$$

*where $L_1$, $L_2$, $L_3$, and $L_4$ are positive constants.*

Next, we theoretically show that a proxy set function $\hat{G}_\theta(S)$ is monotone and $\gamma$-weakly submodular in Theorem 1.

**Theorem 1.** *We define a proxy set function $\hat{G}_\theta(S) \triangleq G_\theta(S) + |S|\sigma$, where $\sigma = 1 + \nu_1 + \nu_2 L_2 + \eta M L_2(L_1 + \eta kN(L_1 L_4 + L_2 L_3))$, $\nu_1 \to 0^+$, and $\nu_2 > 0$ are positive constants. Given Assumption 1, $\hat{G}_\theta(S)$ is monotone and $\gamma$-weakly submodular where $\gamma > \gamma^* = \frac{1}{2\sigma-1}$.*

The proof is in Appendix A.1. We construct a proxy optimization problem in Eq. (5) as follows:

$$\hat{S}^* = \underset{S \subseteq X, |S|/|X| = k}{\arg\max} \hat{G}_\theta(S). \tag{6}$$

According to Theorem 1, a greedy search algorithm [37, 38] can be leveraged to approximately solve the proxy problem in Eq. (6) and it can provide the following optimality guarantee of the optimization problem in Eq. (5).

**Algorithm 1** Robustness-aware Coreset Selection (RCS)

---
1: **Input:** Unlabeled training set $X$, unlabeled validation set $U$, batch size $\beta$, model $g \circ f_\theta$, learning rate for RCS $\eta$, subset fraction $k \in (0, 1]$
2: **Output:** Coreset $S$
3: Initialize $S \leftarrow \emptyset, N \leftarrow |X|$
4: Split entire training set into minibatches $\{B_m\}_{m=1}^{\lceil N/\beta \rceil}$
5: **for** each minibatch $B_m \subset X$ **do**
6:     Compute gradient $q_m \leftarrow \nabla_\theta \mathcal{L}_{\text{ACL}}(B_m; \theta)$
7: **end for**
8: // Conduct greedy search via batch-wise selection
9: **for** $1, \ldots, \lfloor kN/\beta \rfloor$ **do**
10:     Compute gradient $q_U \leftarrow \nabla_\theta \mathcal{L}_{\text{RD}}(U; \theta)$
11:     Initialize $best\_gain = -\infty$
12:     **for** each minibatch $B_m \subset X$ **do**
13:         Compute marginal gain $\hat{G}(B_m|S) \leftarrow \eta q_U^\top q_m$ // refer to Eq. (7)
14:         **if** $\hat{G}(B_m|S) > best\_gain$ **then**
15:             Update $s \leftarrow m, best\_gain \leftarrow \hat{G}(B_m|S)$
16:         **end if**
17:     **end for**
18:     Update $S \leftarrow S \cup B_s, X \leftarrow X \setminus B_s$
19:     Update $\theta \leftarrow \theta - \eta q_s$
20: **end for**

---

**Theorem 2.** *Given a fixed parameter $\theta$, we denote the optimal solution of Eq. (5) as $G_\theta^* = \sup_{S \subseteq X, |S|/|X|=k} G_\theta(S)$. Then, $\hat{S}^*$ in Eq. (6) found via greedy search satisfies*

$$G_\theta(\hat{S}^*) \geq G_\theta^* - (G_\theta^* + kN\sigma) \cdot e^{-\gamma^*}.$$

**Remark.** The proof is in Appendix A.2. Theorem 2 indicates that the greedy search for solving the proxy problem in Eq. (6) can provide a guaranteed lower-bound of the original problem in Eq. (5), which implies that RCS via greedy search can help ACL to obtain the minimized RD and robust representations. We also empirically validate that ACL on the coreset selected by RCS can achieve a lower RD compared with ACL on the randomly selected subset in Figure 5 (in Appendix B.6), which empirically supports that RCS via greedy search is effective in achieving the minimized RD.

**Algorithm of RCS.** Therefore, we use a greedy search algorithm via batch-wise selection for RCS. Algorithm 1 iterates the batch-wise selection $\lfloor k|X|/\beta \rfloor$ times. At each iteration, it finds the minibatch $B$ that has the largest gain $\hat{G}_\theta(B|S)$ based on the parameters updated by the previously selected coreset, and then adds this minibatch into the final coreset. RCS via greedy search needs to calculate the gradient for each minibatch $\lceil N/\beta \rceil$ times (Line 6 in Algorithm 1) and the gradient on the validation set $\lfloor kN/\beta \rfloor$ times (Line 10 in Algorithm 1). In total, RCS needs to calculate the loss gradient $\lceil N/\beta \rceil + \lfloor kN/\beta \rfloor \ll \binom{N}{kN}$ times, which is significantly more efficient than the native solution. We approximate the marginal gain function using the Taylor expansion as follows:

$$\hat{G}_\theta(B|S) \approx \eta \nabla_\theta \mathcal{L}_{\text{RD}}(U; \theta - \eta \nabla_\theta \mathcal{L}_{\text{ACL}}(S; \theta))^\top \nabla_\theta \mathcal{L}_{\text{ACL}}(B; \theta) + \beta\sigma, \tag{7}$$

where $B$ is a minibatch and $\beta$ is the batch size. The derivation of Eq. (7) is in Appendix A.3. It enables us to efficiently calculate the marginal gain for each minibatch (Line 13 in Algorithm 1). We omit this term in Algorithm 1 since $\beta\sigma$ is a constant. Intuitively, RCS selects the data whose training loss gradient and validation loss gradient are of the most similarity. In this way, the model can minimize the RD (validation loss) after updating its parameters by minimizing the ACL loss (training loss) on the coreset selected by RCS, thus helping improve the adversarial robustness of the representation.

### 3.4 Efficient ACL via RCS

We show an efficient ACL procedure via RCS in Algorithm 2. ACL with RCS trains the model on the previously selected coreset for $I \in \mathbb{N}$ epochs, and for every $I$ epochs a new coreset is selected.

---
**Algorithm 2** Efficient ACL via RCS
---
1: **Input:** Unlabeled training set $X$, unlabeled validation set $U$, total training epochs $E$, learning rate $\eta'$, batch size $\beta$, warmup epoch $W$, epoch interval for executing RCS $I$, subset fraction $k$, learning rate for RCS $\eta$
2: **Output:** Adversarially pre-trained feature extractor $f_{\theta'}$
3: Initialize parameters of model $g \circ f_{\theta'}$
4: Initialize training set $S \leftarrow X$
5: // Warmup training for $W$ epochs; Training on the coreset for $(E - W)$ epochs
6: **for** $e = 0$ **to** $(E - 1)$ **do**
7:     **if** $e\%I == 0$ **and** $e \geq W$ **then**
8:         $\theta \leftarrow copy(\theta')$
9:         $S \leftarrow \text{RCS}(X, U, \beta, g \circ f_\theta, \eta, k)$ // by Algorithm 1
10:     **end if**
11:     **for** batch $m = 1, \ldots, \lceil |S|/\beta \rceil$ **do**
12:         Sample a minibatch $B_m$ from $S$
13:         Update $\theta' \leftarrow \theta' - \eta' \nabla_{\theta'} \mathcal{L}_{\text{ACL}}(B_m; \theta')$
14:     **end for**
15: **end for**
---

The pre-training procedure is repeated until the required epoch $E \in \mathbb{N}$ is reached. We provide four tricks with three tricks as follows and one trick that enables efficient RCS on large-scale datasets with limited GPU memory in Appendix B.1.

**Warmup on the entire training set.** We take $W$ epochs to train the model on the entire training set as the warmup. Warmup training enables the model to have a good starting point to provide informative gradients used in RCS. For example, we use $10\%$ of the total training epochs for warmup.

**Last-layer gradients.** It is computationally expensive to compute the gradients over full layers of a deep model due to a tremendous number of parameters in the model. To tackle this issue, we utilize a last-layer gradient approximation, by only considering the loss gradients of the projection head $g$ during RCS.

**Adversarial data approximation.** Calculating adversarial data during CS is extremely time-consuming since generating adversarial data needs to iteratively perform BP $T$ times. We let $T_{\text{ACL}}$ be PGD steps, $\epsilon_{\text{ACL}}$ be the adversarial budget, and $\rho_{\text{ACL}}$ be the step size for PGD during ACL. Similarly, $T_{\text{RCS}}$, $\epsilon_{\text{RCS}}$, and $\rho_{\text{RCS}}$ denote PGD configurations during RCS. To mitigate the above issue, we decrease $T_{\text{RCS}}$ (i.e., $0 < T_{\text{RCS}} \leq T_{\text{ACL}}$) and $\rho_{\text{RCS}} = \frac{\rho_{\text{ACL}} \cdot T_{\text{ACL}}}{T_{\text{RCS}}}$ for efficiently generating adversarial data during CS. Note that the same adversarial budget is used for ACL and RCS, i.e., $\epsilon_{\text{RCS}} = \epsilon_{\text{ACL}}$.

## 4 Experiments

In this section, we first validate that our proposed RCS can significantly accelerate ACL [14] and its variant [17, 15] on various datasets [6, 26] with minor degradation in transferability to downstream tasks. Then, we apply RCS to a large-scale dataset, i.e., ImageNet-1K [1] to demonstrate that RCS enhances the scalability of ACL. Lastly, we demonstrate extensive empirical results. Extensive experimental details are in Appendix B.2.

**Efficient pre-training configurations.** We leverage RCS to speed up ACL [14] and DynACL [17] using ResNet-18 backbone networks. The pre-training settings of ACL and DynACL exactly follow their original paper and we provide the details in Appendix B.2. For the hyperparameters of RCS, we set $\beta = 512$, $\eta = 0.01$, and $T_{\text{RCS}} = 3$. We took $W = 100$ epochs for warmup, and then CS was executed every $I = 20$ epoch. We used different subset fractions $k \in \{0.05, 0.1, 0.2\}$ for CS. The KL divergence was used as the distance function to calculate $\mathcal{L}_{\text{RD}}(\cdot)$ for all the experiments in Section 4. We used the pre-trained weights released in ACL's and DynACL's GitHub as the pre-trained encoder to reproduce their results on the entire set. We repeated all the experiments using different random seeds three times and report the median results to exclude the effect of randomization.

**Finetuning methods.** We used the following three kinds of finetuning methods to evaluate the learned representations: standard linear finetuning (SLF), adversarial linear finetuning (ALF), and

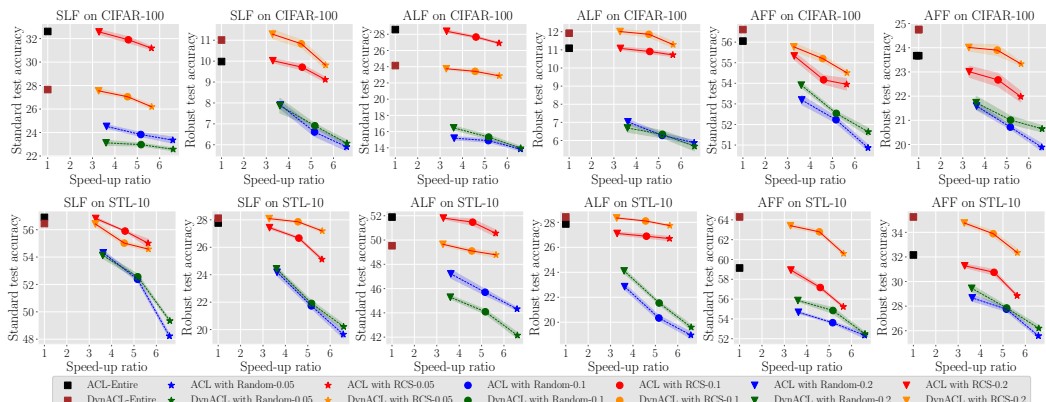

Figure 2: Cross-task adversarial robustness transferability from CIFAR-10 to CIFAR-100 (upper row) and STL-10 (bottom row). The number after the dash line denotes subset fraction $k \in \{0.05, 0.1, 0.2\}$. ACL with RCS and DynACL with RCS correspond to the red and orange solid lines, respectively. ACL with Random and DynACL with Random correspond to the blue and green dotted lines, respectively.

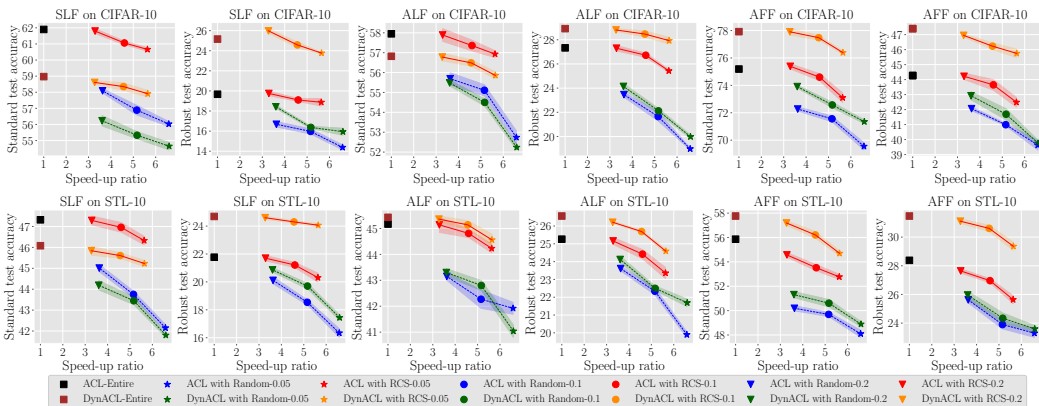

Figure 3: Cross-task adversarial robustness transferability from CIFAR-100 to CIFAR-10 (upper row) and STL-10 (bottom row). The number after the dash line denotes subset fraction $k \in \{0.05, 0.1, 0.2\}$.

adversarial full finetuning (AFF). SLF and ALF will keep the learned encoder frozen and only finetune the linear classifier using natural or adversarial samples, respectively. AFF leverages the pre-trained encoder as weight initialization and trains the whole model using the adversarial data. The finetuning settings exactly follow DynACL [17]. More detials in Appendix B.2.

**Evaluation metrics.** The reported robust test accuracy (dubbed as "RA") is evaluated via AutoAttack (AA) [19] and the standard test accuracy (dubbed as "SA") is evaluated on natural data. In practice, we used the official code of AutoAttack [19] for implementing evaluations. In Appendix B.5, we provide robustness evaluation under more diverse attacks [19, 55, 56].

**Baseline.** We replaced RCS (Line 9 in Algorithm 2) with the random selection strategy (dubbed as "Random") to obtain the coreset $S$ during ACL as the baseline. The implementation of Random exactly follows that in the coreset and data selection library [31].

**Speed-up ratio.** The speed-up ratio denotes the ratio of the running time of *the whole pre-training procedure* on the entire set (dubbed as "Entire") to that of ACL with various CS strategies (i.e., Random and RCS). We report the running time of Entire in Table 3.

### 4.1 Effectiveness of RCS in Efficient ACL

**Cross-task adversarial robustness transferability.** Figures 2 and 3 demonstrate the cross-task adversarial robustness transferability from CIFAR-10 and CIFAR-100 to downstream tasks, respectively. First, we observe that ACL/DynACL with RCS (red/orange lines) always achieves significantly higher robust and standard test accuracy than ACL/DynACL with Random (blue/green lines) among

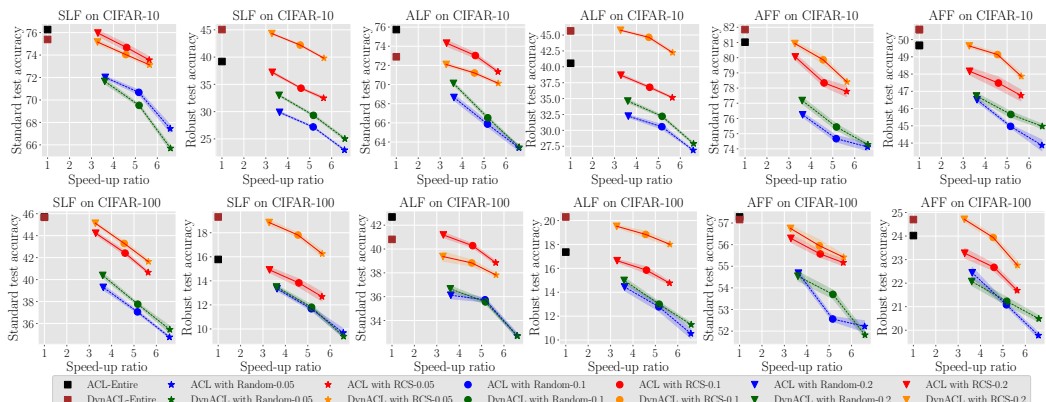

Figure 4: Self-task adversarial robustness transferability evaluated on the CIFAR-10 (left three panels) and CIFAR-100 (right three panels) dataset. The number after the dash line denotes subset fraction $k \in \{0.05, 0.1, 0.2\}$.

various subset fractions while achieving a similar speed-up ratio to Random. Besides, compared with ACL/DynACL on the entire set (black/brown squares), ACL/DynACL with RCS (red/orange lines) can almost maintain robust and standard test accuracy on downstream tasks. Notably, RCS-0.2 almost maintains robust test accuracy on various downstream tasks via various finetuning methods. It validates the effectiveness of our principled method RCS in maintaining robustness transferability while speeding up pre-training.

**Self-task adversarial robustness transferability.** Figure 4 shows the self-task adversarial robustness transferability evaluated on CIFAR-10 and CIFAR-100 datasets where pre-training and finetuning were conducted on the same datasets. In Appendix B.3, we provide the performance on the STL-10 [26] task in Tables 4 and 5. Apparently, ACL/DynACL with RCS (red/orange lines) always yields better performance across tasks than ACL/DynACL with Random (blue/green lines). Besides, RCS (solid lines) can almost maintain robust test accuracy via various finetuning methods compared with Entire (squares) while having a high speed-up ratio. It validates that RCS is effective in efficiently learning robust representations.

## 4.2 Benchmarks on ImageNet-1K

To the best of our knowledge, we are the first to apply ACL [14] on ImageNet-1K efficiently via RCS. As shown in Table 1, ACL on the entire set of ImageNet-1K will need about 650.2 hours using 4 NVIDIA RTX A5000 GPUs, which is extremely time-consuming and unmanageable for us to conduct ACL on the entire set. Thus, we do not provide the results of ACL on the entire set.

Tables 1 and 2 demonstrate the cross-task adversarial robustness transferability of pre-trained WideResNet [57] with width 10 and depth 28 (WRN-28-10) from ImageNet-1K of $32 \times 32$ resolution to CIFAR-10 and CIFAR-100, respectively. Experimental details of pre-training are in Appendix B.2. The two tables demonstrate that ACL with RCS consistently leads to better robust and standard test accuracy on downstream tasks than ACL with Random. Besides, we can observe that ACL with RCS often obtains better robustness transferability compared with standard CL. Surprisingly, ACL with RCS even consumed less running time (111.8 hours) than standard CL (147.4 hours) while achieving better transferability. It validates that RCS effectively enhances the scalability of ACL, and efficient ACL via RCS can yield useful and robust representations.

## 4.3 Extensive Experimental Results

**Ablation study.** We provide the results of RCS using various distance functions in Appendix B.8, RCS with various warmup epochs $W$ in Appendix B.9, RCS with various epoch intervals $I$ for executing RCS in Appendix B.10, RCS with various batch sizes during CS in Appendix B.11, and RCS for another variant AdvCL [15] in Appendix B.12.

**Comparison between ACL with FGSM and ACL with RCS.** To the best of our knowledge, no work has studied how to use FGSM (one-step PGD) [24] to speed up ACL. In Table 7, we demonstrate

Table 1: Cross-task adversarial robustness transferability from ImageNet-1K to CIFAR-10.

| Pre-training | Runing time (hours) | SLF | | ALF | | AFF | |
|---|---|---|---|---|---|---|---|
| | | SA (%) | RA (%) | SA (%) | RA (%) | SA (%) | RA (%) |
| Standard CL | 147.4 | $\mathbf{84.36}_{\pm 0.17}$ | $0.01_{\pm 0.01}$ | $10.00_{\pm 0.00}$ | $10.00_{\pm 0.00}$ | $\mathbf{86.63}_{\pm 0.12}$ | $49.71_{\pm 0.16}$ |
| ACL on entire set | 650.2 | - | - | - | - | - | - |
| ACL with Random | 94.3 | $68.75_{\pm 0.06}$ | $15.89_{\pm 0.06}$ | $59.57_{\pm 0.28}$ | $27.14_{\pm 0.19}$ | $84.75_{\pm 0.18}$ | $50.12_{\pm 0.21}$ |
| ACL with RCS | 111.8 | $70.02_{\pm 0.12}$ | $\mathbf{22.45}_{\pm 0.13}$ | $63.94_{\pm 0.21}$ | $\mathbf{31.13}_{\pm 0.17}$ | $85.23_{\pm 0.23}$ | $\mathbf{52.21}_{\pm 0.14}$ |

Table 2: Cross-task adversarial robustness transferability from ImageNet-1K to CIFAR-100.

| Pre-training | Runing time (hours) | SLF | | ALF | | AFF | |
|---|---|---|---|---|---|---|---|
| | | SA (%) | RA (%) | SA (%) | RA (%) | SA (%) | RA (%) |
| Standard CL | 147.4 | $\mathbf{57.34}_{\pm 0.23}$ | $0.01_{\pm 0.01}$ | $9.32_{\pm 0.01}$ | $0.06_{\pm 0.01}$ | $\mathbf{61.33}_{\pm 0.12}$ | $25.11_{\pm 0.15}$ |
| ACL on entire set | 650.2 | - | - | - | - | - | - |
| ACL with Random | 94.3 | $38.53_{\pm 0.15}$ | $10.50_{\pm 0.13}$ | $28.44_{\pm 0.23}$ | $11.93_{\pm 0.21}$ | $59.63_{\pm 0.33}$ | $25.46_{\pm 0.26}$ |
| ACL with RCS | 111.8 | $40.28_{\pm 0.17}$ | $\mathbf{14.55}_{\pm 0.10}$ | $33.15_{\pm 0.26}$ | $\mathbf{14.89}_{\pm 0.16}$ | $60.25_{\pm 0.18}$ | $\mathbf{28.24}_{\pm 0.13}$ |

that FGSM cannot effectively learn robust representations and consumes more running time than RCS. Therefore, our proposed RCS is more efficient and effective in speeding up ACL than FGSM.

**RCS for efficient supervised robust pre-training.** In Appendix B.13.2, following Hendrycks et al. [58] and Salman et al. [48], we apply RCS to speed up pre-training WRN-28-10 and ResNet-50 via SAT [34] on ImageNet-1K [1] and demonstrate that SAT with RCS can almost maintain transferability. Therefore, RCS can be a unified and effective framework for efficient robust pre-training.

**Comparison between RCS and ACS [33] in speeding up supervised AT including Fast-AT [24], SAT [34], and TRADES [35].** We provide the comparison between RCS and ACS [33] in Appendix B.13.1, which validates that our proposed RCS, without using labels during CS, is more efficient and effective than ACS in terms of speeding up supervised AT [24, 34, 35].

**Analysis of the coreset selected by RCS.** We provide comprehensive quantitative and visualization analyses of the coreset selected by RCS in Appendix B.6. The analyses demonstrate that RCS helps minimize RD and tends to select a coreset that is closer to the entire training set compared to Random, thus helping ACL obtain useful and adversarially robust representations.

# 5   Conclusion

This paper proposed a robustness-aware coreset selection (RCS) framework for accelerating robust pre-training. RCS found an informative subset that helps minimize the representational divergence between natural data and their adversarial counterparts. We theoretically showed that RCS can be efficiently solved by greedy search approximately with an optimality guarantee. RCS does not require label information and is thus applicable to ACL as well as supervised AT. Our experimental results validated that RCS can significantly speed up both ACL and supervised AT while slightly hurting the robustness transferability.

One of the limitations is that RCS still requires spending a particular amount of running time during coreset selection in calculating the loss gradients, although we theoretically propose the greedy-search algorithm with an optimality guarantee to make RCS efficient. We leave how to further improve the efficiency of RCS, such as by leveraging better submodular function optimization methods [59, 60], as the future work.

# Acknowledgements

This research is supported by the National Research Foundation, Singapore under its Strategic Capability Research Centres Funding Initiative, Australian Research Council (ARC) under Award No. DP230101540 and NSF and CSIRO Responsible AI Program under Award No. 2303037. Any opinions, findings and conclusions or recommendations expressed in this material are those of the author(s) and do not reflect the views of National Research Foundation, Singapore.

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

# A  Theoretical Analysis

## A.1   Proof of Theorem 1

**Theorem 1 (restated)**     *We define a proxy set function $\hat{G}_\theta(S) \triangleq G_\theta(S) + |S|\sigma$, where $\sigma = 1 + \nu_1 + \nu_2 L_2 + \eta M L_2(L_1 + \eta k N(L_1 L_4 + L_2 L_3))$, $\nu_1 \to 0^+$, and $\nu_2 > 0$ are positive constants. Given Assumption 1, $\hat{G}_\theta(S)$ is monotone and $\gamma$-weakly submodular where $\gamma > \gamma^* = \frac{1}{2\sigma - 1}$.*

*Proof.*  We first provide **proof of monotonicity of $\hat{G}_\theta(S)$**. We use Taylor expansion two times to convert the marginal gain function $\hat{G}_\theta(x|S)$ as follows,

$$
\hat{G}_\theta(x|S) = \hat{G}_\theta(S \cup \{x\}) - \hat{G}_\theta(S)
$$

$$
= -\mathcal{L}_{\mathrm{RD}}(U; \theta - \eta\frac{\partial}{\partial\theta}\mathcal{L}_{\mathrm{ACL}}(S; \theta) - \eta\frac{\partial}{\partial\theta}\mathcal{L}_{\mathrm{ACL}}(\{x\}; \theta)) + \mathcal{L}_{\mathrm{RD}}(U; \theta - \eta\frac{\partial}{\partial\theta}\mathcal{L}_{\mathrm{ACL}}(S; \theta)) + \sigma
$$

$$
\xlongequal{\text{Taylor expansion}} \eta\frac{\partial}{\partial\theta}\mathcal{L}_{\mathrm{RD}}(U; \theta - \eta\frac{\partial}{\partial\theta}\mathcal{L}_{\mathrm{ACL}}(S; \theta))^\top \frac{\partial}{\partial\theta}\mathcal{L}_{\mathrm{ACL}}(\{e\}; \theta) + \xi_1(\theta) + \sigma
$$

$$
\xlongequal{\text{Taylor expansion}} \eta\frac{\partial}{\partial\theta}\left[ \mathcal{L}_{\mathrm{RD}}(U; \theta) - \eta\frac{\partial}{\partial\theta}\mathcal{L}_{\mathrm{RD}}(U; \theta)^\top \frac{\partial}{\partial\theta}\mathcal{L}_{\mathrm{ACL}}(S; \theta) + \xi_2(\theta) \right]^\top \frac{\partial}{\partial\theta}\mathcal{L}_{\mathrm{ACL}}(\{x\}; \theta)
$$

$$
+ \xi_1(\theta) + \sigma
$$

$$
= \eta\frac{\partial}{\partial\theta}\mathcal{L}_{\mathrm{RD}}(U; \theta)^\top \frac{\partial}{\partial\theta}\mathcal{L}_{\mathrm{ACL}}(\{x\}; \theta) - \eta^2 \frac{\partial}{\partial\theta}\mathcal{L}_{\mathrm{RD}}(U; \theta)^\top \frac{\partial^2}{\partial\theta}\mathcal{L}_{\mathrm{ACL}}(S; \theta)\frac{\partial}{\partial\theta}\mathcal{L}_{\mathrm{ACL}}(\{x\}; \theta)
$$

$$
- \eta^2 \frac{\partial}{\partial\theta}\mathcal{L}_{\mathrm{ACL}}(S; \theta)^\top \frac{\partial^2}{\partial\theta}\mathcal{L}_{\mathrm{RD}}(U; \theta)\frac{\partial}{\partial\theta}\mathcal{L}_{\mathrm{ACL}}(\{x\}; \theta) + \frac{\partial}{\partial\theta}\xi_2(\theta)^\top \frac{\partial}{\partial\theta}\mathcal{L}_{\mathrm{ACL}}(\{x\}; \theta)
$$

$$
+ \xi_1(\theta) + \sigma, \tag{8}
$$

where $\xi_1(\theta) \to 0$ and $\xi_2(\theta) \to 0$ are the remainder terms of Taylor series. Recall that $\mathcal{L}_{\mathrm{RD}}(U; \theta) = \sum_{x_i \in U} \ell_{\mathrm{RD}}(x_i; \theta)$, $\mathcal{L}_{\mathrm{ACL}}(S; \theta) = \sum_{x_i \in S} \ell_{\mathrm{ACL}}(x_i; \theta)$, $|U| = M$, and $|X| = N$. According to Assumption 1, we have the following results :

$$
\|\frac{\partial}{\partial\theta}\mathcal{L}_{\mathrm{RD}}(U; \theta)\| \le |U|L_1 = ML_1; \tag{9}
$$

$$
\|\frac{\partial^2}{\partial\theta}\mathcal{L}_{\mathrm{RD}}(U; \theta)\| \le |U|L_3 = ML_3; \tag{10}
$$

$$
\|\frac{\partial}{\partial\theta}\mathcal{L}_{\mathrm{ACL}}(S; \theta)\| \le |S|L_2 \le k|X|L_2 = kNL_2; \tag{11}
$$

$$
\|\frac{\partial^2}{\partial\theta}\mathcal{L}_{\mathrm{ACL}}(S; \theta)\| \le |S|L_4 \le k|X|L_4 = kNL_4. \tag{12}
$$

We assume that $\|\xi_1(\theta)\| \le \nu_1$ and $\|\frac{\partial}{\partial\theta}\xi_2(\theta)\| \le \nu_2$ where $\nu_1 \to 0^+$ and $\nu_2 > 0$ are two positive constant. We can transform Eq. (8) as follows,

$$
\hat{G}_\theta(x|S) \ge -\|\hat{G}_\theta(x|S)\|
$$

$$
\ge -\eta\|\frac{\partial}{\partial\theta}\mathcal{L}_{\mathrm{RD}}(U; \theta)\|\|\frac{\partial}{\partial\theta}\mathcal{L}_{\mathrm{ACL}}(\{x\}; \theta)\| - \eta^2\|\frac{\partial}{\partial\theta}\mathcal{L}_{\mathrm{RD}}(U; \theta)\|\|\frac{\partial^2}{\partial\theta}\mathcal{L}_{\mathrm{ACL}}(S; \theta)\|\|\frac{\partial}{\partial\theta}\mathcal{L}_{\mathrm{ACL}}(\{x\}; \theta)\|
$$

$$
- \eta^2\|\frac{\partial}{\partial\theta}\mathcal{L}_{\mathrm{ACL}}(S; \theta)\|\|\frac{\partial^2}{\partial\theta}\mathcal{L}_{\mathrm{RD}}(U; \theta)\|\|\frac{\partial}{\partial\theta}\mathcal{L}_{\mathrm{ACL}}(\{e\}; \theta)\| - \|\frac{\partial}{\partial\theta}\xi_2(\theta)\|\|\frac{\partial}{\partial\theta}\mathcal{L}_{\mathrm{ACL}}(\{x\}; \theta)\|
$$

$$
- \|\xi_1(\theta)\| + \sigma
$$

$$
\ge -\eta ML_1 L_2 - \eta^2 ML_1 kNL_4 L_2 - \eta^2 kNL_2 ML_3 L_2 - \nu_2 L_2 - \nu_1 + \sigma
$$

$$
\ge -\eta ML_2(L_1 + \eta kN(L_1 L_4 + L_2 L_3)) - \nu_2 L_2 - \nu_1 + \sigma.
$$

Since $\sigma = \eta ML_2(L_1 + \eta kN(L_1 L_4 + L_2 L_3)) + \nu_1 + \nu_2 L_2 + 1$, we can obtain that

$$
\hat{G}_\theta(x|S) \ge 1. \tag{13}
$$

Eq. (13) shows that $\hat{G}_\theta(S)$ is monotone, i.e., $\hat{G}_\theta(x|S) > 0$.

Then, we provide **proof of $\gamma$-weakly submodularity of $\hat{G}_\theta(S)$**. The sketch of proof is that we first prove $\hat{G}_\theta(S)$ is $\alpha$-submodular. Then, according to Lemma 1, we can conclude that $\hat{G}_\theta(S)$ is also $\gamma$-weakly submodular.

**Lemma 1** (Proposition 4 of Gatmiry and Gomez-Rodriguez [38]). *The set function $G(S) : 2^X \to \mathbb{R}$ that is $\alpha$-submodular is $\gamma$-weakly submodular with the submodularity ratio $\gamma \geq 1 - \alpha$.*

The following is the definition of $\alpha$-submodularity.

**Definition 2** ($\alpha$-submodularity [38]). *A function is called $\alpha$-submodular if the marginal gain of adding an element $x$ to set $A$ is $1 - \alpha$ times greater than or equals to the gain of adding an element $x$ to set $B$ where $A \subseteq B$. i.e.,*

$$\underset{A,B|A\subseteq B}{\forall} G_\theta(x|A) \geq (1-\alpha)G_\theta(x|B).$$

According to Eq. (13), we have obtained the lower bound of the marginal gain

$$\hat{G}_\theta(x|S) \geq 1 = \underline{\hat{G}}_\theta(S). \tag{14}$$

We can obtain the upper bounder of marginal gain based on Eq. (8) as follows,

$$\hat{G}_\theta(x|S) \leq \eta\|\frac{\partial}{\partial_\theta}\mathcal{L}_{\text{RD}}(U;\theta)\|\|\frac{\partial}{\partial_\theta}\mathcal{L}_{\text{ACL}}(\{x\};\theta)\| + \eta^2\|\frac{\partial}{\partial_\theta}\mathcal{L}_{\text{RD}}(U;\theta)\|\|\frac{\partial^2}{\partial_\theta}\mathcal{L}_{\text{ACL}}(S;\theta)\|\|\frac{\partial}{\partial_\theta}\mathcal{L}_{\text{ACL}}(\{x\};\theta)\|$$

$$+ \eta^2\|\frac{\partial}{\partial_\theta}\mathcal{L}_{\text{ACL}}(S;\theta)\|\|\frac{\partial^2}{\partial_\theta}\mathcal{L}_{\text{RD}}(U;\theta)\|\|\frac{\partial}{\partial_\theta}\mathcal{L}_{\text{ACL}}(\{x\};\theta)\| + \|\frac{\partial}{\partial_\theta}\xi_2(\theta)\|\|\frac{\partial}{\partial_\theta}\mathcal{L}_{\text{ACL}}(\{x\};\theta)\|$$

$$+ \|\xi_1(\theta)\| + \sigma$$

$$\leq \eta M L_1 L_2 + \eta^2 M L_1 k N L_4 L_2 + \eta^2 k N L_2 M L_3 L_2 + \nu_2 L_2 + \nu_1 + \sigma$$

$$\leq \eta M L_2 (L_1 + \eta k N(L_1 L_4 + L_2 L_3)) + \nu_2 L_2 + \nu_1 + \sigma.$$

Since $\sigma = \eta M L_2(L_1 + \eta k N(L_1 L_4 + L_2 L_3)) + \nu_2 L_2 + \nu_1 + 1$, we can obtain that

$$\hat{G}_\theta(x|S) \leq 2\eta M L_2(L_1 + \eta k N(L_1 L_4 + L_2 L_3)) + 2\nu_1 + 2\nu_2 L_2 + 1 = 2\sigma - 1 = \overline{\hat{G}}_\theta(S). \tag{15}$$

Based on the lower bound (Eq. (14)) and the upper bound (Eq. (15)) of the marginal gain function, we can obtain

$$\frac{\hat{G}_\theta(x|A)}{\hat{G}_\theta(x|B)} \geq \frac{\underline{\hat{G}}_\theta(S)}{\overline{\hat{G}}_\theta(S)} = \frac{1}{2\eta M L_2(L_1 + \eta k N(L_1 L_4 + L_2 L_3)) + 2\nu_1 + 2\nu_2 L_2 + 1}$$

$$= \frac{1}{2\sigma - 1} = 1 - \alpha^* \in (0, 1). \tag{16}$$

Therefore, we have proved $\hat{G}_\theta(S)$ is $\alpha^*$-submodular. According to Lemma 1, $\hat{G}_\theta(S)$ is $\gamma$-weakly submodular where $\gamma > \gamma^* = \frac{1}{2\sigma - 1}$.

$\square$

## A.2 Proof of Theorem 2

**Theorem 2 (restated)** *Given a fixed parameter $\theta$, we denote the optimal solution of Eq. (5) as $G_\theta^* = \sup_{S\subseteq X, |S|/|X|=k} G_\theta(S)$. Then, $\hat{S}^*$ in Eq. (6) found via greedy search satisfies*

$$G_\theta(\hat{S}^*) \geq G_\theta^* - (G_\theta^* + kN\sigma) \cdot e^{-\gamma^*}.$$

*Proof.* We provide the lemma that describes the optimality guarantee of the greedy search algorithm in the problem of maximizing a monotone $\alpha$-approximate submodular function subject to cardinality constraints.

**Lemma 2** ([37, 38]). *Given a monotone and $\alpha$-approximate submodular function $G(S)$, the greedy search algorithm achieves a $1 - e^{-(1-\alpha)}$ approximation factor for the problem of maximizing $G(S)$ subject to cardinality constraints.*

Note that according to Theorem 1, $\hat{G}_\theta(S)$ is a monotone and $\alpha^*$-approximate submodular function. Therefore, according to Lemma 2, we have

$$\hat{G}_\theta(\hat{S}^*) \geq (1 - e^{-\gamma^*})\hat{G}_\theta^*, \tag{17}$$

where $\hat{G}_\theta^* = \sup_{S \subseteq X, |S|/|X|=k} \hat{G}_\theta(S)$ and $\hat{S}^*$ is found via greedy search. Since $\hat{G}_\theta(S) = G_\theta(S) + |S|\sigma$, we have

$$\hat{G}_\theta^* = \sup_{S \subseteq X, |S|/|X|=k} (G_\theta(S) + |S|\sigma) = \sup_{S \subseteq X, |S|/|X|=k} G_\theta(S) + k|X|\sigma = G_\theta^* + kN\sigma. \tag{18}$$

Recall that $\hat{G}_\theta(\hat{S}^*) = G_\theta(\hat{S}^*) + kN\sigma$. Therefore, by transforming Eq. (17), we can obtain

$$G_\theta(\hat{S}^*) \geq G_\theta^* - (G_\theta^* + kN\sigma) \cdot e^{-\gamma^*}. \tag{19}$$

$\square$

### A.3 Derivation of Eq. (7)

Here, we provide the details of deriving Eq. (7) using Taylor expansion as follows:

$$\hat{G}_\theta(B|S) = \hat{G}_\theta(S \cup B) - \hat{G}_\theta(S) \tag{20}$$
$$= G_\theta(S \cup B) + |S \cup B|\sigma - (G_\theta(S) + |S|\sigma) \tag{21}$$
$$= G_\theta(S \cup B) - G_\theta(S) + \beta\sigma \tag{22}$$
$$= -\mathcal{L}_{\mathrm{RD}}(U; \theta - \eta\nabla_\theta\mathcal{L}_{\mathrm{ACL}}(S \cup B; \theta)) + \mathcal{L}_{\mathrm{RD}}(U; \theta - \eta\nabla_\theta\mathcal{L}_{\mathrm{ACL}}(S; \theta)) + \beta\sigma \tag{23}$$
$$= -\mathcal{L}_{\mathrm{RD}}(U; \theta - \eta\nabla_\theta\mathcal{L}_{\mathrm{ACL}}(S; \theta) - \eta\nabla_\theta\mathcal{L}_{\mathrm{ACL}}(B; \theta))$$
$$\quad + \mathcal{L}_{\mathrm{RD}}(U; \theta - \eta\nabla_\theta\mathcal{L}_{\mathrm{ACL}}(S; \theta)) + \beta\sigma \tag{24}$$
$$= -\mathcal{L}_{\mathrm{RD}}(U; \theta_S - \eta\nabla_\theta\mathcal{L}_{\mathrm{ACL}}(B; \theta)) + \mathcal{L}_{\mathrm{RD}}(U; \theta_S) + \beta\sigma \tag{25}$$
$$\approx -\left(\mathcal{L}_{\mathrm{RD}}(U; \theta_S) - \eta\nabla_\theta\mathcal{L}_{\mathrm{RD}}(U; \theta_S)^\top\nabla_\theta\mathcal{L}_{\mathrm{ACL}}(B; \theta) + \xi\right) + \mathcal{L}_{\mathrm{RD}}(U; \theta_S) + \beta\sigma \tag{26}$$
$$\approx \eta\nabla_\theta\mathcal{L}_{\mathrm{RD}}(U; \theta_S)^\top\nabla_\theta\mathcal{L}_{\mathrm{ACL}}(B; \theta) + \beta\sigma, \tag{27}$$

where $B$ is a minibatch whose batch size is $\beta \in \mathbb{N}$, $S$ is the subset, and $S$ and $B$ are disjoint subsets. Eq. (20) holds according to the definition of marginal gain in Definition 1. Eq. (21) holds according to the definition of the proxy set function $\hat{G}_\theta(S)$ in Theorem 1. Eq. (22) holds because $S$ and $B$ are disjoint. Eq. (23) holds according to the definition of set function in Eq. (4). Eq. (24) holds by splitting the ACL loss on $S \cup B$ into the sum of the ACL loss on $S$ and the ACL loss on $B$. Eq. (25) holds by letting $\theta_S = \theta - \eta\nabla_\theta\mathcal{L}_{\mathrm{ACL}}(S; \theta)$. In Eq. (26), we transform $\mathcal{L}_{\mathrm{RD}}(U; \theta_S - \eta\nabla_\theta\mathcal{L}_{\mathrm{ACL}}(B; \theta))$ via Taylor expansion where $\xi$ is the remainder. Eq. (27) holds by omitting the remainder. By replacing $\theta_S$ with $\theta - \eta\nabla_\theta\mathcal{L}_{\mathrm{ACL}}(S; \theta)$ in Eq. (27), we can obtain Eq. (7).

## B Extensive Experimental Details and Results

### B.1 An Extra Trick to Enable Efficient RCS on Large-Scale Datasets with Limited GPU Memory

RCS on large-scale datasets such as ImageNet-1K [1] needs to calculate the loss gradient for each minibatch of training data as the first step (Line 5–7 in Algorithm 1). Then, the greedy search is conducted to iteratively select the minibatch that has the largest marginal gain and add this minibatch into the final coreset (Line 9–20 in Algorithm 1). Saving the gradients of all the minibatches on GPU needs a large amount of GPU memory. When the GPU memory is limited, there exists an issue that we are unable to save all the gradients on GPU.

A direct solution is to transfer the loss gradients from GPU to CPU and save all the gradients on CPU. When we need to use the loss gradient of a minibatch, we can transfer the gradient from CPU to GPU

for calculation. Unexpectedly, the transition between CPU and GPU still consumes a large amount of time in practice. Thus, we do not take this solution.

To solve this issue, we split the entire training set into several training subsets. Practically, we take every 100 minibatches of training data as a training subset. Then, we conduct RCS on each training subset respectively and collect the coresets from each training subset together as the final coresets for robust pre-training. In this way, we can enable RCS to efficiently search for the coresets from large-scale datasets with limited GPU memory. We apply this trick to the experiments regarding ACL on ImageNet-1K in Section 4.2 and SAT on ImageNet-1K in Section 4.3.

## B.2 Complementary Experimental Details

We conducted all experiments on Python 3.8.8 (PyTorch 1.13) with 4 NVIDIA RTX A5000 GPUs (CUDA 11.6).

**Experimental details of ACL [14].** Following Jiang et al. [14], we leveraged the dual BN [61], where one BN is used for the standard branch of the feature extractor and the other BN is used for the adversarial branch, for the implementation of ResNet-18 and WRN-28-10 backbone models.

As for ACL using ResNet-18 on CIFAR-10 and CIFAR-100, we pre-trained ResNet-18 models using SGD for 1000 epochs with an initial learning rate 5.0 and a cosine annealing schedule [62]. During pre-training, we set the adversarial budget $\epsilon_{\text{ACL}} = 8/255$, the step size $\alpha_{\text{ACL}} = 2/255$, and the PGD step $T_{\text{ACL}} = 5$ for generating adversarial training data.

As for ACL using WRN-28-10 on ImageNet-1K, we pre-trained WRN-28-10 on ImageNet-1K of $32 \times 32$ resolution via ACL [14] with RCS and Random, respectively. We set $\beta = 512, T_{\text{ACL}} = 5, \epsilon_{\text{ACL}} = 8/255, \rho_{\text{ACL}} = 2/255, T_{\text{RCS}} = 1$. The temperature parameter $t$ was set as 0.1, following Chen et al. [13]. The model was trained using SGD for 200 epochs with an initial learning rate 0.6 and a cosine annealing scheduler. We set warmup epoch $W = 200$, and then CS was executed every $I = 10$ epochs. We pre-trained WRN-28-10 with the subset fraction k = 0.05 and then finetuned it on CIFAR10 and CIFAR-100.

**Experimental details of DynACL [17].** The pre-training settings of DynACL [17] followed ACL [14], except for the strength of data augmentation and the hyperparameter $\omega$. We denote the strength of data augmentation and the hyperparameter at epoch $e$ as $\mu_e$ and $\omega_e$ respectively, where

$$\mu_e = 1 - \lfloor \frac{e}{K} \rfloor \cdot \frac{K}{E}, \quad e \in \{0, 1, \ldots, E - 1\},$$
$$\omega_e = \nu \cdot (1 - \mu_e), \quad e \in \{0, 1, \ldots, E - 1\},$$

in which the decay period $K = 50$, the reweighting rate $\nu = 2/3$, the total training epoch $E = 1000$. In our implementation of DynACL, we only take the dynamic strategy and do not take the trick of the stop gradient operation and the momentum encode [63, 64].

**Experimental details of finetuning.** As for linear finetuning (SLF and ALF), we fixed the parameter of the encode and only trained the linear classifier using SGD with an initial learning rate 0.01 divided by 10 at Epoch 10 and 20 for 25 epochs. As for AFF, we trained the whole model for 25 epochs using SGD. The adversarial training data is generated via PGD with 10 PGD steps, adversarial budget $8/255$, and step size $2/255$. Practically, we exactly used the finetuning code provided in the GitHub of DynACL [17].

**Baseline models that are pre-trained on the entire set.** As for the pre-trained ResNet-18 on the entire set of CIFAR-10 and CIFAR-100 via ACL and DynACL, we used their released pre-trained weights in their GitHub[3][4][5][6] to reproduce their results.

As for the pre-trained WRN-28-10 on the entire set of ImageNet-1K, due to the extremely long pre-training time of ACL on the entire set (as shown in Table 3), we do not provide the results of ACL on the full set of ImageNet-1K. We only provide the results of ACL on the coreset in Table 1 and 2.

---

[3]Link of pre-trained weights via ACL on CIFAR-10.

[4]Link of pre-trained weights via ACL on CIFAR-100.

[5]Link of pre-trained weights via DynACL on CIFAR-10.

[6]Link of pre-trained weights via DynACL on CIFAR-100.

Table 3: Running time (hours) on the entire training set of various models via ACL, DynACL, SAT, and TRADES.

| Pre-training on the entire set | ResNet-18 on CIFAR-10 | ResNet-50 on ImageNet-1K | WRN-28-10 on ImageNet-1K |
|---|---|---|---|
| ACL [14] | 42.8 | - | 650.2 |
| DynACL [17] | 42.9 | - | - |
| SAT [34] | 5.2 | 286.1 | 341.7 |
| TRADES [35] | 5.5 | - | - |

Table 4: Self-task adversarial robustness transferability of ACL with RCS evaluated on the STL-10 dataset. The number after the dash line denotes subset fraction $k \in \{0.05, 0.1, 0.2\}$.

| Pre-training | Runing time (hours) | SLF | |
|---|---|---|---|
| | | SA (%) | RA (%) |
| ACL-Entire | 74.4 | 71.16 | 33.21 |
| ACL with Random-0.05 | 10.9 | 62.86 | 22.00 |
| ACL with RCS-0.05 | **13.1** | **66.77** | **26.99** |
| ACL with Random-0.1 | 13.9 | 64.54 | 24.16 |
| ACL with RCS-0.1 | **16.4** | **68.17** | **28.66** |
| ACL with Random-0.2 | 19.4 | 66.61 | 27.09 |
| ACL with RCS-0.2 | **22.6** | **70.26** | **30.09** |

Table 5: Self-task adversarial robustness transferability of DynACL with RCS evaluated on the STL-10 dataset. The number after the dash line denotes subset fraction $k \in \{0.05, 0.1, 0.2\}$.

| Pre-training | Runing time (hours) | SLF | |
|---|---|---|---|
| | | SA (%) | RA (%) |
| DynACL-Entire | 74.7 | 69.63 | 46.51 |
| DynACL with Random-0.05 | 10.9 | 55.05 | 27.38 |
| DynACL with RCS-0.05 | **13.2** | **63.75** | **37.01** |
| DynACL with Random-0.1 | 14.1 | 58.77 | 30.72 |
| DynACL with RCS-0.1 | **16.6** | **67.42** | **40.88** |
| DynACL with Random-0.2 | 19.7 | 62.79 | 34.94 |
| DynACL with RCS-0.2 | **22.9** | **69.14** | **43.27** |

As for pre-training via SAT [34] on the entire set of ImageNet-1K, we use WRN-28-10 pre-trained on ImageNet-1K of $32 \times 32$ resolution provided by Hendrycks et al. [58] and ResNet-50 ($\epsilon = 4.0$) pre-trained on the entire set of ImageNet-1K of $224 \times 224$ resolution provided by Salman et al. [48].

Table 3 demonstrates the approximated training time on the entire set of various models via ACL [14], DynACL [17], SAT [34], and TRADES [35], respectively. All the running time was evaluated on NVIDIA RTX A5000 GPUs. We used 1 GPU for robust pre-training on CIFAR-10 and CIFAR-100, and 4 GPUs for pre-training on ImageNet-1K.

### B.3 Self-Task Adversarial Robustness Transferability Evaluated on the STL-10 Dataset

Following the setting of [17], we applied our proposed method RCS to speed up ACL and DynACL on STL-10 [26] and report the results in Tables 4 and 5. The results validate that RCS is effective in speeding up both ACL and DynACL while being able to learn useful robust representations.

### B.4 Experimental Details of Figure 1

In Figure 1, we report the cross-task robustness transferability of pre-trained ResNet-18 via ACL with RCS (subset fraction $k = 0.1$) from CIFAR-10 to CIFAR-100 and STL-10 via SLF. The experimental settings are the same as Section 4.1.

### B.5 Robustness Evaluated under Various Attacks

In this subsection, we report the adversarial robustness evaluated under three strong white-box attacks (APGD-CE [19], APGD-DLR [19] and FAB [55]) and one strong black-box attack (i.e., Square Attack [56]). We evaluate the cross-task adversarial robustness transferability from CIFAR-100 to STL-10 via ALF under various attacks and report the results in Table 6. The results validate that RCS can consistently improve robust test accuracy over various adversaries compared with Random and almost maintain the robustness transferability while significantly speeding up the pre-training.

Table 6: Adversarial robustness evaluated under various attacks including APGD-CE [19], APGD-DLR [19], FAB [55], and Square Attack [56]. The number after the dash line denotes subset fraction $k = 0.1$.

| Pre-training | Running time (hours) | APGD-CE (%) | APGD-DLR (%) | FAB (%) | Square Attack (%) |
|---|---|---|---|---|---|
| ACL | 42.8 | 29.70 | 25.55 | 24.60 | 28.61 |
| ACL with Random-0.1 | 8.3 | 27.29 | 23.70 | 22.69 | 26.39 |
| ACL with RCS-0.1 | **9.3** | **29.10** | **26.77** | **25.62** | **29.15** |
| DynACL | 42.9 | 31.35 | 27.41 | 26.76 | 28.61 |
| DynACL with Random-0.1 | 8.3 | 26.56 | 22.44 | 21.44 | 25.39 |
| DynACL with RCS-0.1 | **9.4** | **29.23** | **26.89** | **25.99** | **29.50** |

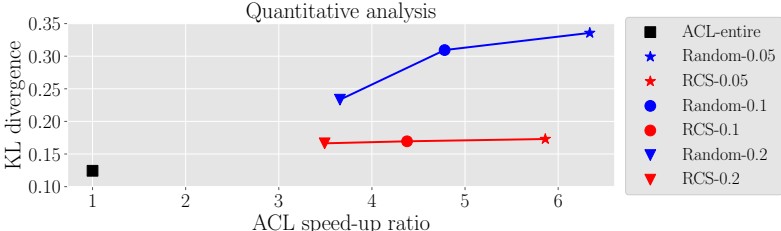

Figure 5: Average value of the KL divergence between natural test data and its adversarial variant.

## B.6 Analysis of the Coreset Selected by RCS

**Quantitative analysis of the coreset selected by RCS.** Here, we provide comprehensive quantitative analyses of the coreset selected by RCS in Figures 5 and 6 to help understand the effectiveness of RCS in maintaining transferability.

In Figure 5, we demonstrate the average value of the KL divergence between natural data point $x$ and its PGD-20 adversarial variant on the CIFAR-10 test set (i.e., $\ell_{RD}(x)$). Specifically, we collected the average value of the KL divergence between natural data point $x$ and its PGD-20 adversarial variant on the CIFAR-10 test set (i.e., $\ell_{RD}(x)$) and show it in the leftmost panel of Figure 5. It validates that RCS helps minimize RD, thus helping ACL obtain useful and adversarially robust representations. It validates that RCS helps minimize RD, thus helping ACL obtain useful and adversarially robust representations.

In Figure 6, we demonstrate the imbalance ratio and the maximum mean discrepancy (MMD) between the coreset and the whole training set. Note that the imbalance ratio is the ratio of the sample size of the largest majority class and that of the smallest minority class. MMD is a classical measurement of the distance between two distributions. The left panel of Figure 6 shows that the corset selected by RCS is almost class-balanced since the imbalance ratio of RCS is slightly higher than 1.0. The right panel of Figure 6 shows that RCS yields a lower MMD between the entire training set and the selected coreset compared to Random. Therefore, our quantitative analysis demonstrates that RCS selects a coreset that is closer to the entire training set than Random, thus helping to maintain transferability.

**Visualization analysis of the coreset selected by RCS.** We count the frequency of each training sample in the CIFAR-10 dataset being selected into the coreset. In Figure 7, we visualize the top-5 most-frequently selected (MFS) training data and the top-5 least-frequently selected (LFS) training data. Figure 7 shows that, compared to LFS data, MFS data are images whose backgrounds are more complex and are more difficult to be differentiated from the subject. Recent work [65, 43, 66] has shown that exempting the representations from the nuisance style factors such as the background factor can improve the robustness against distribution shifts. RCS prefers to select images of complex backgrounds to help the model learn representations that are independent of the background factors, thus helping maintain robustness against adversarial perturbations.

## B.7 Comparison Between ACL with FGSM and ACL with RCS

To the best of our knowledge, previously no one has studied how to incorporate FGSM [24, 18] with ACL to speed up ACL with FGSM. Previous studies [24, 25] only leveraged the trick for speeding up supervised AT where labels are required. Here, we take the primitive step to study the performance of FGSM in efficient ACL.

We demonstrate the comparison between ACL with FGSM and ACL with RCS on the CIFAR-10 task, which validates that our proposed RCS is more efficient and effective than FGSM [24].

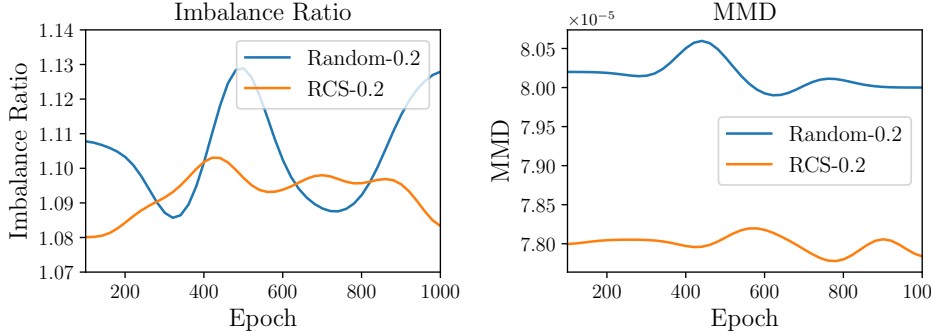

Figure 6: We show the imbalance ratio of the coreset (left panel) and the maximum mean discrepancy (MMD) between coreset and the full training set (right panel) at each epoch. We conducted ACL with RCS and ACL with Random on the CIFAR-10 dataset using $k = 0.2$. The coreset is selected every $I = 20$ epoch. We used a Gaussian filter ($\sigma = 3$) to smooth the line.

Most-Frequently Selected (MFS) Data

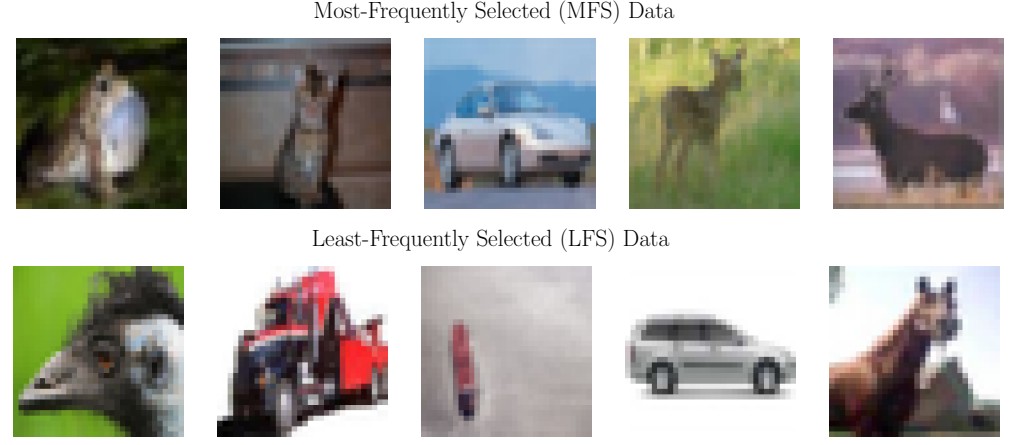

Least-Frequently Selected (LFS) Data

Figure 7: We show the most-frequently selected data (upper panel) and the least-frequently selected data from the CIFAR-10 training data.

Table 7: The self-task adversarial robustness transferability achieved by ACL with FGSM and ACL with RCS on the CIFAR-10 dataset. The number after the dash line denotes the subset fraction $k = 0.2$.

| Pre-training | Runing time (hours) | SLF | | ALF | |
|---|---|---|---|---|---|
| | | SA (%) | RA (%) | SA (%) | RA (%) |
| ACL-Entire | 42.8 | 78.87 | 39.19 | 75.74 | 40.56 |
| ACL with FGSM | 21.7 | 86.70 | 0.00 | 60.04 | 0.11 |
| ACL with RCS-0.2 | **13.0** | **75.96** | **37.21** | **74.32** | **38.68** |
| DynACL-Entire | 42.9 | 75.39 | 45.05 | 72.90 | 45.65 |
| DynACL with FGSM | 21.8 | 83.89 | 1.02 | 70.83 | 3.27 |
| DynACL with RCS-0.2 | **13.1** | **75.16** | **44.32** | **72.10** | **45.75** |

We conducted pre-training on CIFAR-10 using ResNet-18 via both ACL and DynACL on the entire set with FGSM (denoted as "ACL with FGSM" and "DynACL with FGSM" respectively) and then conducted the linear evaluation on CIFAR-10. Following Fast-AT [24], we used one-step PGD with the adversarial budget as 8/255 and step size as 10/255 for FGSM. Other training and finetuning configurations (e.g., optimizer and learning rate scheduler) are the same as in Section 4.1.

In Table 7, we show the comparison between RCS (subset fraction $k = 0.2$) and FGSM in speeding up ACL. We can observe that FGSM cannot effectively learn useful robust representations. In addition, RCS even consumes less running time than FGSM. Therefore, our proposed method RCS is more efficient and effective than FGSM [24] in speeding up ACL.

Table 8: Self-task adversarial robustness transferability achieved by ACL with RCS using various distance functions evaluated on the CIFAR-10 dataset. The number after the dash line denotes subset fraction $k \in \{0.05, 0.1, 0.2\}$.

| Pre-training | Distance function | Runing time (hours) | SLF | |
| --- | --- | --- | --- | --- |
| | | | SA (%) | RA (%) |
| ACL-Entire | - | 42.8 | 78.87 | 39.19 |
| ACL with Random-0.05 | - | 6.5 | 67.45 | 22.96 |
| ACL with RCS-0.05 | JS | 7.7 | 72.39 | 32.43 |
| ACL with RCS-0.05 | OT | 7.9 | 72.49 | 32.36 |
| ACL with RCS-0.05 | KL | **7.6** | **72.56** | **32.49** |
| ACL with Random-0.1 | - | 8.3 | 70.68 | 27.19 |
| ACL with RCS-0.1 | JS | 9.4 | 74.43 | 34.23 |
| ACL with RCS-0.1 | OT | 9.5 | 74.39 | 34.26 |
| ACL with RCS-0.1 | KL | **9.3** | **74.67** | **34.30** |
| ACL with Random-0.2 | - | 11.8 | 72.01 | 29.87 |
| ACL with RCS-0.2 | JS | 13.2 | 75.75 | 37.18 |
| ACL with RCS-0.2 | OT | 13.3 | 75.59 | 37.06 |
| ACL with RCS-0.2 | KL | **13.0** | **75.96** | **37.21** |

Table 9: Self-task adversarial robustness transferability achieved by ACL with RCS using various warmup epochs evaluated on the CIFAR-10 dataset.

| Pre-training | Warmup epoch | Runing time (hours) | SLF | |
| --- | --- | --- | --- | --- |
| | | | SA (%) | RA (%) |
| ACL-Entire | - | 42.8 | 78.87 | 39.19 |
| ACL with Random | 100 | 11.8 | 72.01 | 29.87 |
| ACL with RCS | 100 | **13.0** | **75.96** | **37.21** |
| ACL with Random | 200 | 15.2 | 73.38 | 30.59 |
| ACL with RCS | 200 | **16.3** | **76.43** | **38.33** |
| ACL with Random | 300 | 18.7 | 73.91 | 31.62 |
| ACL with RCS | 300 | **19.6** | **78.18** | **39.01** |

## B.8 Efficient ACL via RCS with Various Distance Functions $\Phi$

We pre-trained ACL with RCS using various distance functions on CIFAR-10 and then report the performance via linear evaluation on CIFAR-10 in Table 8. Other training settings exactly keep the same as Section 4.1, except for the distance functions.

It validates that efficient ACL via RCS with various distance functions can consistently obtain better transferability compared with Random. Especially, RCS with KL divergence often achieves better transferability to downstream tasks among different distance functions. Therefore, we use KL divergence for RCS in all the experiments in the main paper.

## B.9 Efficient ACL via RCS with Various Warmup Epochs $W$

This section provides the results of RCS with different warmup epochs $W \in \{100, 200, 300\}$ for efficient ACL. We set the subset fraction $k$ as 0.2. Other training settings follow Section 4.1. Table 9 shows that as the warmup epoch of RCS increases, the performance evaluated on the CIFAR-10 task becomes better while consuming more running time. Besides, RCS still consistently achieves better performance than Random, which validates the effectiveness of RCS in speeding up ACL while maintaining robustness transferability.

## B.10 Efficient ACL via RCS with Various Epoch Intervals $I$

We pre-trained ResNet-18 on CIFAR-10 via ACL with RCS using $I \in \{10, 20, 50\}$, $k = 0.2$, and $W = 100$. Then, we evaluate the performance in the CIFAR-10 task via SLF. Table 10 shows that a larger $I$ leads to more pre-training time since the frequency of conducting coreset selection becomes larger. Meanwhile, a larger $I$ can lead to higher robust and standard test accuracy in downstream tasks. This is because a larger $I$ enables the coreset to be updated more timely to adapt to the latest state of the model and select data points that can help the model improve its robustness.

Table 10: Impact of $I$ for RCS evaluated on the CIFAR-10 task.

| $I$ | Runing time (hours) | SLF | |
|---|---|---|---|
| | | SA (%) | RA (%) |
| 10 | 15.6 | **76.31** | **38.17** |
| 20 | 13.0 | 75.96 | 37.21 |
| 50 | **12.2** | 75.87 | 35.54 |

Table 11: Impact of batch size for coreset selection evaluated on the CIFAR-10 task.

| Pre-training | Batch size for RCS | Runing time (hours) | SLF | |
|---|---|---|---|---|
| | | | SA (%) | RA (%) |
| ACL-Entire | - | 42.8 | 78.87 | 39.19 |
| ACL with Random | - | 11.8 | 72.01 | 29.87 |
| ACL with RCS | 64 | 13.6 | 76.21 | 37.48 |
| ACL with RCS | 128 | 13.3 | 76.15 | 37.41 |
| ACL with RCS | 256 | 13.1 | 75.89 | 37.17 |
| ACL with RCS | 512 | **13.0** | **75.96** | **37.21** |

Table 12: Compatibility with AdvCL [15] evaluated on the CIFAR-10 task via SLF.

| Pre-training | Runing time (hours) | SLF | |
|---|---|---|---|
| | | SA (%) | RA (%) |
| AdvCL-Entire | 57.8 | 80.89 | 42.36 |
| AdvCL with Random | 11.0 | 73.67 | 33.61 |
| AdvCL with RCS | **13.5** | **77.93** | **38.89** |

## B.11 Efficient ACL via RCS with Various Batch Sizes

In this subsection, we show the impact of the batch size during coreset selection. We trained ResNet-18 via ACL with RCS on CIFAR-10 and then linearly finetuned ResNet-18 models on CIFAR-10. The batch size for RCS is selected from $\{64, 128, 256, 512\}$ and the batch size for ACL keeps as 512. The subset fraction keeps as 0.2. Other training settings exactly follow Section 4.1. We report the standard and robust test accuracy in Table 11.

We can find that as the batch size for RCS decreases, the running time becomes larger. It is because there is a larger number of batches needed to calculate the loss gradients during RCS when the batch size becomes smaller. Besides, we observe that the test accuracy on the downstream tasks seems to gradually increase as the batch size decreases. Especially, ACL with RCS using 64 batch size gains consistent improvement compared with ACL with RCS using 512 batch size, which indicates that a smaller batch size for RCS is useful to improve the performance but consumes more running time. Therefore, there could be a trade-off between the running time and the transferability. We leave how to further improve the efficiency and effectiveness in maintaining the transferability of RCS as the future work.

## B.12 RCS for Accelerating Another Variant AdvCL [15]

Fan et al. [15] proposed a variant of ACL method, called "AdvCL", that leverages a standardly pre-trained model on ImageNet-1K to generate pseudo-labels for CIFAR-10 training data via $K$-means clustering. Based on ACL [14], the loss function of AdvCL is composed of a weighted sum of the adversarial contrastive loss and an ensemble of the CE loss between the adversarial data and its pseudo label over different choices of cluster number. By simply replacing the loss function of the ACL $\mathcal{L}_{\mathrm{ACL}}(\cdot)$ with that of AdvCL in Algorithm 1 and 2, we can apply RCS for efficient AdvCL.

We pre-trained ResNet-18 via AdvCL with RCS on CIFAR-10 using SGD for 1000 epochs with an initial learning rate 0.5 and a cosine annealing scheduler. We set $\beta = 512$, $T_{\mathrm{ACL}} = 5$, $\epsilon_{\mathrm{ACL}} = 8/255$, $\rho_{\mathrm{ACL}} = 2/255$, and $T_{\mathrm{RCS}} = 3$. The pre-training settings of AdvCL exactly follow Fan et al. [15]. We take the same configuration of RCS as that of RCS for ACL in Section 4.1, i.e., warmup epoch $W = 100$, the epoch interval for conducting RCS $I = 20$, and the subset fraction $k = 0.1$. Then, we evaluate the performance on the CIFAR-10 task via SLF.

Table 12 shows that RCS can speed up AdvCL [15] as well. Besides, RCS is a principled method that helps AdvCL to obtain effective robust representations since AdvCL with RCS always achieves a higher test accuracy compared with AdvCL with Random. Therefore, the experimental results

---

**Algorithm 3** RCS for Supervised AT

---

1: **Input:** Labeled training set $D$, validation set $U$, batch size $\beta$, model $g \circ f_\theta$, learning rate for RCS $\eta$, subset fraction $k \in (0,1]$, loss function of supervised AT method $\mathcal{L}_{supervised}(\cdot)$
2: **Output:** Coreset $S$
3: Initialize $S \leftarrow \emptyset, N \leftarrow |D|$
4: Split entire set into minibatches $\{B_m\}_{m=1}^{\lceil N/\beta \rceil}$
5: **for** each minibatch $B_m \subset D$ **do**
6:     Compute gradient $q_m \leftarrow \nabla_\theta \mathcal{L}_{supervised}(B_m; \theta)$
7: **end for**
8: // Conduct greedy search via batch-wise selection
9: **for** $1, \ldots, \lfloor kN/\beta \rfloor$ **do**
10:     Compute gradient $q_U \leftarrow \nabla_\theta \mathcal{L}_{\mathrm{RD}}(U; \theta)$
11:     Initialize $best\_gain = -\infty$
12:     **for** each minibatch $B_m \subset D$ **do**
13:         Compute marginal gain $\hat{G}(B_m|S) \leftarrow \eta q_U^\top q_m$
14:         **if** $\hat{G}(B_m|S) > best\_gain$ **then**
15:             Update $s \leftarrow m, best\_gain \leftarrow \hat{G}(B_m|S)$
16:         **end if**
17:     **end for**
18:     Update $S \leftarrow S \cup B_s, D \leftarrow D \setminus B_s$
19:     Update $\theta \leftarrow \theta - \eta q_s$
20: **end for**

---

---

**Algorithm 4** Efficient Supervised AT via RCS

---

1: **Input:** Labeled training set $D$, validation set $U$, total training epochs $E$, learning rate $\eta'$, batch size $\beta$, warmup epoch $W$, epoch interval for executing RCS $I$, subset fraction $k$, learning rate for RCS $\eta$, loss function of supervised AT method $\mathcal{L}_{supervised}(\cdot)$
2: **Output:** robust pre-trained feature extractor $f'_\theta$
3: Initialize parameters of model $g \circ f'_\theta$
4: Initialize training set $S \leftarrow D$
5: // Warmup training for $W$ epochs; Training on the coreset for $(E-W)$ epochs
6: **for** $e = 0$ **to** $(E-1)$ **do**
7:     **if** $e\%I == 0$ **and** $e \geq W$ **then**
8:         $\theta \leftarrow copy(\theta')$
9:         $S \leftarrow \mathrm{RCS}(D, U, \beta, g \circ f_\theta, \eta, k, \mathcal{L}_{supervised}(\cdot))$ // by Algorithm 3
10:     **end if**
11:     **for** batch $m = 1, \ldots, \lceil |S|/\beta \rceil$ **do**
12:         Sample a minibatch $B_m$ from $S$
13:         Update $\theta' \leftarrow \theta' - \eta' \nabla_{\theta'} \mathcal{L}_{supervised}(B_m; \theta')$
14:     **end for**
15: **end for**

---

validate that our proposed RCS can be a unified and effective framework for accelerating ACL as well as its variants [15, 17].

### B.13   RCS for Accelerating Supervised AT

In this section, we first provide the algorithm of RCS for supervised AT including fast adversarial training (Fast-AT) [24], free adversarial training (Free-AT) [25], standard adversarial training (SAT) [34] and TRADES [35]. Then, we show the comparison between RCS and ACS [33] in speeding up supervised AT on CIFAR-10 and the effectiveness of RCS in efficient SAT on ImageNet-1K [1].

**Algorithm of RCS for supervised AT.**   Prior to introducing the algorithm of RCS for SAT and TRADES, we first provide the preliminaries of SAT and TRADES.

Given a labeled training set $D = \{(x_i, y_i)\}_{i=1}^N$, where data $x_i \in \mathcal{X}$ and label $y_i \in \mathcal{Y} = \{0, 1, \ldots, C-1\}$, a feature extractor $f_\theta : \mathcal{X} \to \mathcal{Z}$, a randomly initialized classifier $g : \mathcal{Z} \to \mathbb{R}^C$,

Table 13: Comparison between Fast-AT [24] with ACS [33] and Fast-AT with RCS on CIFAR-10.

| Training method | Running time (minutes) | SA (%) | RA under PGD-20 (%) | RA under AutoAttack (%) |
|---|---|---|---|---|
| Fast-AT on the entire set | 25.0 | 86.20 | 45.80 | 41.04 |
| Fast-AT with ACS | 12.8 | 82.71 | 45.71 | 40.92 |
| Fast-AT with RCS | **12.0** | **83.47** | **45.89** | **41.06** |

Table 14: Compatibility with Free-AT [25].

| Training method | Running time (minutes) | SA (%) | RA under PGD-20 (%) | RA under AutoAttack (%) |
|---|---|---|---|---|
| Free-AT on the entire set | 116.5 | 84.18 | 49.05 | 45.14 |
| Fast-AT with RCS | **60.6** | **82.39** | **49.24** | **45.15** |

the loss function of SAT is

$$\mathcal{L}_{\text{SAT}}(D;\theta) = \sum_{i=1}^{N} \left\{ \max_{\tilde{x}_i \in \mathcal{B}_\epsilon[x_i]} \ell(g \circ f_\theta(\tilde{x}_i), y_i) \right\}, \tag{28}$$

where $\ell$ is the Cross-Entropy (CE) loss and $\tilde{x}_i$ is adversarial training data generated by PGD within the $\epsilon$-ball centered at $x_i$.

The loss function of TRADES is

$$\mathcal{L}_{\text{TRADES}}(D;\theta) = \sum_{i=1}^{N} \left\{ \ell(g \circ f_\theta(x_i), y_i) + c \cdot \max_{\tilde{x}_i \in \mathcal{B}_\epsilon[x_i]} KL(g \circ f_\theta(\tilde{x}_i), g \circ f_\theta(x_i)) \right\}, \tag{29}$$

where $\ell$ is the CE loss, $KL(\cdot, \cdot)$ is the KL divergence, $c > 0$ is a trade-off parameter, and $\tilde{x}_i$ is adversarial training data generated by PGD within the $\epsilon$-ball centered at $x_i$. We set $c = 6$, following Zhang et al. [35]. Note that the parameters of $g$ are updated during supervised AT. Here we omit the parameters of $g$ since we only use the parameters of the feature extractor $f_\theta$ in downstream tasks.

The RCS problem for supervised AT is formulated as follows:

$$S^* = \underset{S \subseteq D, |S|/|D| = k}{\arg\max} -\mathcal{L}_{\text{RD}}(U; \theta - \eta \nabla_\theta \mathcal{L}_{supervised}(S;\theta)), \tag{30}$$

in which we replace the ACL loss $\mathcal{L}_{\text{ACL}}(\cdot)$ in Eq. (5) with the supervised AT loss $\mathcal{L}_{supervised}(\cdot)$ (e.g., $\mathcal{L}_{\text{SAT}}(\cdot)$ and $\mathcal{L}_{\text{TRADES}}(\cdot)$). Due to that $\mathcal{L}_{\text{RD}}(\cdot)$ only needs data and does not need any label, RCS is applicable to supervised AT, no matter if the validation set is unlabeled or labeled. By leveraging greedy search, we show the algorithm of RCS for supervised AT in Algorithm 3 and efficient supervised AT via RCS in Algorithm 4.

### B.13.1 Comparison Between RCS and ACS [33] in Speeding Up Supervised AT including Fast-AT [24], SAT [34] and TRADES [35]

**Comparison between Fast-AT with RCS and Fast-AT [24] with ACS [33].** We conducted Fast-AT, Fast-AT with ACS, and Fast-AT with RCS on CIFAR-10. The experimental setting of Fast-AT and Fast-AT with ACS exactly follows that in Section 4.3 of Dolatabadi et al. [33]. That is, we trained ResNet-18 on CIFAR-10 via Fast-AT ($\epsilon = 8/255, \alpha = 10/255$) using SGD with 0.9 momentum for 60 epochs with the initial learning rate of 0.1 and divided by 10 at Epoch 40 and 50. As for the RCS for speeding up Fast-AT, we set the subset fraction as $k = 0.5$, warmup epoch as $W = 10$, and epoch interval for executing RCS $I = 10$. We report the standard test accuracy and robust test accuracy evaluated by PGD-20 ($\epsilon = 8/255, \alpha = 8/2550$) and AutoAttack in Table 13. All experiments are conducted on one RTX A5000 GPU. Table 13 validate that RCS, without using labels, is more efficient and effective in speeding up Fast-AT.

**Compatibility with Free-AT [25].** Further, we conducted Free-AT [25] and Free-AT with RCS on CIFAR-10. We trained ResNet-18 on CIFAR-10 via Free-AT ($\epsilon = 8/255$) using SGD with 0.9 momentum for 60 epochs with the initial learning rate of 0.01 and divided by 10 at Epoch 40 and 50. We keep the configurations of RCS for speeding up Free-AT the same as above. We report the standard test accuracy and robust test accuracy evaluated by PGD-20 and AutoAttack in Table 14.

Therefore, RCS, without using label information, can further speed up both Fast-AT [24] and Free-AT [25] while almost maintaining the adversarial robustness. In addition, RCS without using labels is more efficient than ACS while achieving similar adversarial robustness compared with ACS. Therefore, it validates the effectiveness of RCS in efficient supervised AT.

Table 15: Comparison between SAT [34] with ACS [33] and SAT with RCS on CIFAR-10. Here, "RA" stands for the robust test accuracy under PGD-20 attacks following the setting of Dolatabadi et al. [33]. The number after the dash line denotes subset fraction $k \in \{0.05, 0.1, 0.2\}$.

| Training method | Running time (minutes) | SA (%) | RA (%) |
|---|---|---|---|
| SAT on the entire set | 314 | 82.49 | 52.11 |
| SAT with Random-0.05 | 92 | 69.46 | 32.66 |
| SAT with ACS-0.05 | 142 | 78.44 | 49.31 |
| SAT with RCS-0.05 | **106** | **78.83** | **49.46** |
| SAT with Random-0.1 | 100 | 73.69 | 35.42 |
| SAT with ACS-0.1 | 151 | 79.08 | 50.43 |
| SAT with RCS-0.1 | **115** | **80.17** | **50.54** |
| SAT with Random-0.2 | 119 | 76.89 | 35.72 |
| SAT with ACS-0.2 | 172 | 80.71 | 50.80 |
| SAT with RCS-0.2 | **131** | **81.89** | **50.87** |

Table 16: Comparison between TRADES [35] with ACS [33] and TRADES with RCS on CIFAR-10. Here, "RA" stands for the robust test accuracy under PGD-20 attacks following the setting of Dolatabadi et al. [33]. The number after the dash line denotes subset fraction $k \in \{0.05, 0.1, 0.2\}$.

| Training method | Running time (minutes) | SA (%) | RA (%) |
|---|---|---|---|
| TRADES on the entire set | 332 | 82.72 | 54.35 |
| TRADES with Random-0.05 | 111 | 74.56 | 40.12 |
| TRADES with ACS-0.05 | 152 | 77.23 | 49.59 |
| TRADES with RCS-0.05 | **123** | **77.52** | **50.28** |
| TRADES with Random-0.1 | 122 | 75.80 | 42.19 |
| TRADES with ACS-0.1 | 168 | 78.00 | 50.46 |
| TRADES with RCS-0.1 | **136** | **78.59** | **50.83** |
| TRADES with Random-0.2 | 146 | 77.41 | 44.07 |
| TRADES with ACS-0.2 | 193 | 79.19 | 51.52 |
| TRADES with RCS-0.2 | **161** | **79.82** | **51.84** |

**Efficient SAT [34] and TRADES 29 via RCS on CIFAR-10.** We trained ResNet-18 on CIFAR-10 via SAT [34] using SGD with 0.9 momentum for 120 epochs with the initial learning rate of 0.1 divided by 10 at Epoch 30 and 60. For PGD configurations during SAT, we set $\beta = 128$, $T_{\text{SAT}} = 10$, $\epsilon_{\text{SAT}} = 8/255$, $\rho_{\text{SAT}} = 2/255$. During RCS, we set $T_{\text{RCS}} = 3$. We took $W = 30$ epochs for warmup, and then CS was conducted every $I = 10$ epochs with different subset fractions $k \in \{0.05, 0.1, 0.2\}$.

We trained ResNet-18 on CIFAR-10 via TRADES using SGD with 0.9 momentum for 100 epochs with the initial learning rate of 0.1 divided by 10 at Epoch 60 and 90. For PGD configurations during TRADES, we set $T_{\text{TRADES}} = 10$, $\epsilon_{\text{TRADES}} = 8/255$, $\rho_{\text{TRADES}} = 2/255$. Other training settings follow SAT.

Here, we used ACS [33] and random selection (dubbed as "Random") as the baseline. The implementation of ACS exactly follows the public GitHub of ACS [33]. We use the same subset fraction and warmup epoch for both ACS and Random as RCS.

In Tables 15 and 16, we used the robust test accuracy of adversarial data generated via PGD-20 to evaluate adversarial robustness. It shows that RCS always achieves higher robust and standard test accuracy while consuming running time compared with Random and ACS [33] among various subset fractions. It validates the effectiveness of our proposed RCS in maintaining the adversarial robustness of supervised adversarial training.

### B.13.2 RCS for Efficient Supervised Robust Pre-Training on ImageNet-1K

**Effecitveness of RCS in speeding up STA on ImageNet-1K [1] and maintaining robustness transferability [58] of SAT.** Hendrycks et al. [58] pointed out that the ImageNet-1K adversarially pre-trained models can improve the adversarial robustness on downstream tasks.

Following Hendrycks et al. [58], we pre-trained WRN-28-10 on ImageNet-1K of $32 \times 32$ resolution using SAT [34]. The model was trained using SGD with 0.9 momentum for 100 epochs with an initial learning rate of 0.1 and a cosine decay scheduler. For PGD configurations during SAT, we

Table 17: Cross-task adversarial robustness transferability of adversarially pre-trained WRN-28-10 from ImageNet-1K to CIFAR-10. Here, "RA" stands for robust test accuracy under PGD-20 attacks following the setting of Hendrycks et al. [58]. The number after the dash line denotes subset fraction $k \in \{0.05, 0.1, 0.2\}$.

| Pre-training | Runing time (hours) | ALF | | AFF | |
|---|---|---|---|---|---|
| | | SA (%) | RA (%) | SA (%) | RA (%) |
| Standard training on entire set | 66.7 | 10.12 | 10.04 | 84.73 | 51.91 |
| SAT [58] on entire set | 341.7 | 85.90 | 50.89 | 89.35 | 59.68 |
| SAT with Random-0.05 | 53.6 | 69.59 | 31.58 | 85.55 | 53.53 |
| SAT with RCS-0.05 | **68.6** | **79.72** | **44.44** | **87.99** | **56.87** |
| SAT with Random-0.1 | 70.2 | 73.28 | 33.86 | 86.78 | 54.95 |
| SAT with RCS-0.1 | **81.9** | **81.92** | **45.10** | **88.87** | **57.69** |
| SAT with Random-0.2 | 103.4 | 75.46 | 39.62 | 86.64 | 56.46 |
| SAT with RCS-0.2 | **121.9** | **83.94** | **46.88** | **89.54** | **58.13** |

Table 18: Cross-task adversarial robustness transferability of adversarially pre-trained WRN-28-10 from ImageNet-1K to CIFAR-100. Here, "RA" stands for robust test accuracy under PGD-20 attacks following the setting of Hendrycks et al. [58]. The number after the dash line denotes subset fraction $k \in \{0.05, 0.1, 0.2\}$.

| Pre-training | Runing time (hours) | ALF | | AFF | |
|---|---|---|---|---|---|
| | | SA (%) | RA (%) | SA (%) | RA (%) |
| Standard training on entire set | 66.7 | 11.05 | 0.00 | 54.81 | 27.81 |
| SAT [58] on entire set | 341.7 | 63.35 | 32.08 | 61.56 | 35.64 |
| SAT with Random-0.05 | 53.6 | 41.1 | 19.69 | 61.12 | 31.03 |
| SAT with RCS-0.05 | **68.6** | **58.31** | **27.95** | **64.95** | **33.74** |
| SAT with Random-0.1 | 70.2 | 50.57 | 20.99 | 63.6 | 31.97 |
| SAT with RCS-0.1 | **81.9** | **59.92** | **28.11** | **66.12** | **34.41** |
| SAT with Random-0.2 | 103.4 | 51.05 | 24.19 | 61.97 | 33.58 |
| SAT with RCS-0.2 | **121.9** | **61.90** | **29.01** | **67.40** | **35.91** |

set $\beta = 256$, $T_{\text{SAT}} = 10$, $\epsilon_{\text{SAT}} = 8/255$, $\rho_{\text{SAT}} = 2/255$. We set $T_{\text{RCS}} = 3$ during RCS. We took $W = 10$ epochs for warmup, and then CS was executed every $W = 10$ epochs. We adversarially pre-train WRN-28-10 with different subset fractions $k \in \{0.05, 0.1, 0.2\}$ and then conducted adversarial finetuning on CIFAR-10 and CIFAR-100.

To evaluate the cross-task adversarial robustness transferability of pre-trained WRN-28-10 models via SAT, we adversarially trained the pre-trained models on CIFAR-10 and CIFAR-100 for 30 epochs using SGD with the initial learning rate 0.01 and a cosine annealing scheduler. The learning objective of robust finetuning exactly follows $\mathcal{L}_{\text{Robust}}(\cdot)$ shown in Section 2.2. Note that we used the adversarially pre-trained WRN-28-10 on the entire set released in the GitHub of Hendrycks et al. [58] to reproduce the results of the baseline.

The results shown in Table 17 and 18 validate that RCS can significantly accelerate SAT on ImageNet-1K. Besides, SAT with RCS always obtains much better robustness transferability than SAT with Random and consistently achieves better transferability than standard pre-training as well. Therefore, it validates that RCS can be an effective method to speed up supervised robust pre-training as well.

**Effecitveness of RCS in speeding up SAT on ImageNet-1K [1] and maintaining standard transferability [48] of SAT.** Salman et al. [48] empirically discovered that adversarially pre-trained models on ImageNet-1K can yield higher standard test accuracy on downstream tasks than standardly pre-trained models. Following Salman et al. [48], we pre-trained ResNet-50 models on ImageNet-1K of $224 \times 224$ resolution using SAT [34]. The model was trained using SGD with 0.9 momentum for 90 epochs with an initial learning rate 0.1 divided by 10 at Epoch 30 and 60. For PGD configurations during SAT, we set $\beta = 256$, $T_{\text{SAT}} = 3$, $\epsilon_{\text{SAT}} = 4/255$, $\rho_{\text{SAT}} = \frac{3\epsilon_{\text{SAT}}}{T_{\text{SAT}}}$. We set $T_{\text{RCS}} = 1$ during RCS. We took $W = 10$ epochs for warmup, and then CS was conducted every $I = 10$ epochs. We adversarially pre-trained ResNet-50 with different subset fractions $k \in \{0.05, 0.1, 0.2\}$ and then conducted standard finetuning on downstream tasks i.e., CIFAR-10 and CIFAR-100.

To evaluate the cross-task standard transferability of pre-trained ResNet-50 via SAT on ImageNet-1K of $224 \times 224$ resolution, we conducted standard linear finetuning (SLF) and standard full finetuning (SFF) on downstream tasks (i.e., CIFAR-10 and CIFAR-100) for 150 epochs using SGD. As for SFF,

Table 19: Cross-task standard transferability [48] of adversarially pre-trained ResNet-50 from ImageNet-1K to CIFAR-10 and CIFAR-100, respectively. We report the standard test accuracy (%) via standard linear finetuning (SLF) and standard full finetuning (SFF). The number after the dash line denotes subset fraction $k \in \{0.05, 0.1, 0.2\}$.

| Pre-training | Runing time (hours) | CIFAR-10 | | CIFAR-100 | |
|---|---|---|---|---|---|
| | | SLF | SFF | SLF | SFF |
| Standard training [48] on entire set | - | 78.84 | 97.41 | 57.09 | 84.21 |
| SAT [48] on entire set | 286.1 | 93.53 | 98.09 | 77.29 | 86.99 |
| Fast-AT [24] on entire set | 10.4 | 90.91 | 97.54 | 73.35 | 83.33 |
| SAT with Random-0.05 | 38.7 | 85.72 | 95.27 | 69.29 | 82.34 |
| SAT with RCS-0.05 | **48.2** | **92.68** | **97.65** | **75.35** | **84.71** |
| SAT with Random-0.1 | 45.8 | 87.14 | 95.60 | 71.23 | 83.62 |
| SAT with RCS-0.1 | **55.4** | **92.92** | **97.82** | **75.41** | **85.22** |
| SAT with Random-0.2 | 70.3 | 87.69 | 96.10 | 72.05 | 84.14 |
| SAT with RCS-0.2 | **79.8** | **93.48** | **98.06** | **76.39** | **85.44** |

we finetuned all the parameters of the encoder. We set the initial learning rate of 0.001 for standard full finetuning and 0.01 for standard partial finetuning. The learning rate is divided by 10 at Epoch 50 and 100. Here, we use the same data augmentation as Salman et al. [48] to resize the images of CIFAR-10 and CIFAR-100 to $224 \times 224$. Note that we used the standardly pre-trained ResNet-50 on the entire set and adversarially pre-trained ResNet-50 on the entire set released in the GitHub of Salman et al. [48] to reproduce the results of baselines.

Table 19 shows that RCS substantially accelerates SAT on ImageNet-1K while consistently achieving higher standard test accuracy on downstream tasks than standard pre-training. Besides, SAT with RCS always obtains much higher standard test accuracy than SAT with Random. It validates that RCS is an effective method for efficient supervised AT.

In addition, we also provide a comparison between the standard transferability of Fast-AT [24] and SAT with RCS. We downloaded the pre-trained ResNet-50 vis Fast-AT released in the GitHub of Wong et al. [24]. Although SAT with RCS consumes more running time than Fast-AT, SAT with RCS obtains significantly higher standard test accuracy on downstream tasks, which validates the effectiveness of RCS in maintaining the transferability of supervised robust pre-training methods.

Therefore, our proposed RCS can be a unified and effective framework for speeding up both supervised and unsupervised robust pre-training while maintaining the transferability of pre-trained models.

## C Possible Negative Societal Impacts

Our paper aims to speed up robust pre-training by decreasing the number of training data, which decreases the running time of pre-training and should be friendly to the environment. However, to implement all the experiments in this project, we have conducted the pre-training procedure at least 264 times and the finetuning procedure at least 2376 times, which definitely emitted a bunch of $CO_2$ and consumed a large amount of electricity. Therefore, we hope that our project can help improve the efficiency of robust pre-training and make pre-training more environmental-friendly in the future.

