# OpenReview forum: "Efficient Adversarial Contrastive Learning via Robustness-Aware Coreset Selection"
_NeurIPS.cc/2023/Conference — NeurIPS 2023 spotlight_

### Official Review · Reviewer_HkKZ · 2023-07-06

**Soundness:** 4 excellent
**Presentation:** 4 excellent
**Contribution:** 3 good
**Rating:** 7
**Confidence:** 4

**Summary:**

This paper focuses on the problem of efficient adversarial contrastive learning, the authors propose Robustness-Aware Coreset selection to speed up ACL, and according to the theoretical analysis and experimental results, the proposed framework is effective while does not hurt the performance.

**Strengths:**

1. The topic is interesting and the proposed method is simple yet effective;
2. The experiments are well-designed and solid. Also, several runs are performed on each task, making it more convincing;
3. The writing is easy to follow;
4. The proofs are well-written and sound.

**Weaknesses:**

1. According to the experimental results, it still takes much time to run the proposed algorithm. While the authors discussed it in the limitations, it might a challenge for it to be put into practical use;
2. Figure 2,3,4 are a bit hard to read, might need some re-work;


**Questions:**

Please refer to the previous section

**Limitations:**

The authors discussed limitations in the paper appendix.

---

> ### Author Rebuttal · Authors · 2023-08-04
>
> Many thanks for your supportive and constructive comments! Please find our replies below.
>
> > 1. [Reply to W1] Thanks for pointing out this challenge!
>
> We conjecture that applying better submodular function optimization methods to solve the objective function of our proposed RCS can further improve efficiency. For example, [1] has shown the greedy algorithm can be further accelerated through lazy evaluations. [2] proposes to efficiently approximate the set function with some relative error when the set function itself is difficult and computationally expensive to calculate. Meanwhile, [2] shows that most methods of maximizing submodular functions are robust against such errors.
>
> Besides, our work is orthogonal to large batch optimization [3] in practice. We conjecture that incorporating our proposed method with the large batch optimization methods would be practical for efficiently learning robust representations using large-scale datasets and large models.
>
> > 2. [Reply to W2] Thanks for the constructive suggestion! We will provide more explanations in the captions of the figures in revisions.
>
> We will add clarifications in the captions: ACL with RCS and DynACL with RCS correspond to the red and orange solid lines, respectively. ACL with Random and DynACL with Random correspond to the blue and green dotted lines, respectively.
>
> *References*
>
>
> [1] Krause, Andreas, and Daniel Golovin. "Submodular function maximization." Tractability 3.71-104 (2014): 3.\
> [2] Golovin, Daniel, and Andreas Krause. "Adaptive submodularity: Theory and applications in active learning and stochastic optimization." Journal of Artificial Intelligence Research 42 (2011): 427-486.\
> [3] Large Batch Optimization for Deep Learning: Training BERT in 76 minutes, You et al., ICLR 2020.

---

> > ### Comment · Reviewer_HkKZ · 2023-08-21
> >
> > I thank the authors for the reponses. I will keep my score.

---

### Official Review · Reviewer_ZwZm · 2023-07-06

**Soundness:** 4 excellent
**Presentation:** 4 excellent
**Contribution:** 4 excellent
**Rating:** 6
**Confidence:** 4

**Summary:**

This paper introduces a robustness-aware coreset selection (RCS) method without requiring label information to speed up adversarial contrastive learning. RCS selects an informative training subset that minimizes the representational divergence (RD) between adversarial and natural data. Theoretically, the authors prove a greedy search algorithm can solve a proxy problem and provide the lower bound of the solution. Empirically, comprehensive experimental results demonstrate RCS can significantly speed up ACL while maintaining robustness transferability. To my knowledge, it is the first effort to apply ACL on the large-scale datasets ImageNet-1K and obtain effective robust representations.

**Strengths:**

(1) The proposed method is reasonable and novel. The authors propose to select subsets guided by the RD between natural data and its adversarial variant. RCS does not need label annotations while the existing related work requires labels during coreset selection, which supports the originality of the proposed method. This paper is the first to obtain robust representations by ACL pre-training on ImageNet-1K efficiently via RCS. I think this paper has adequately cited the related work.

(2) The submission is technically sound. The claims are well supported by theoretical analyses and experimental results. Theoretically, the authors prove solving a proxy problem efficiently via the greedy search can guarantee the optimality of the solution for the original problem. Empirically, the authors apply RCS to speed up ACL and its variant DynACL on various datasets and show that RCS can maintain natural and robust test accuracy on various downstream tasks. Besides, the authors provide extensive results that validate RCS can be applied to accelerate supervised adversarial training including Madry’s and TRADES on CIFAR-10 and ImageNet-1K.

(3) Self-supervised robust pre-training can provide improved robustness transferability without requiring label annotations. However, due to the computational prohibition, ACL methods have not been applied to large-scale datasets previously. This paper solves this important issue and enabled ACL to be conducted on large-scale datasets, tackling a very practical meaningful chanllenge.

(4) This paper is well-organized and well-written. The reviewer can easily follow most of the content.


**Weaknesses:**

Minor comment - although the authors propose to use the greedy search algorithm to efficiently search the coreset, it still needs to consume extra time for CS during robust pre-training. How to further improve the efficiency of ACL and maintain its effectiveness should be an interesting future direction.

**Questions:**

See weakness.

**Limitations:**

The limitations and the possible negative societal impacts of this submission have been adequately discussed by the authors in the appendix.

---

> ### Author Rebuttal · Authors · 2023-08-04
>
> Many thanks for your positive and constructive comments!
>
> > [Reply to Weakness] We believe this is an interesting future direction!
>
> We conjecture that applying a better submodular function optimization method to solve the objective function of our proposed RCS can further improve efficiency. For example, [1] has shown the greedy algorithm can be further accelerated through lazy evaluations. [2] proposes to efficiently approximate the set function with some relative error when the set function itself is difficult and computationally expensive to calculate. Meanwhile, [2] shows that most methods of maximizing submodular functions are robust against such errors.
>
> *References*
>
> [1] Krause, Andreas, and Daniel Golovin. "Submodular function maximization." Tractability 3.71-104 (2014): 3.\
> [2] Golovin, Daniel, and Andreas Krause. "Adaptive submodularity: Theory and applications in active learning and stochastic optimization." Journal of Artificial Intelligence Research 42 (2011): 427-486.

---

### Official Review · Reviewer_LfDi · 2023-07-07

**Soundness:** 3 good
**Presentation:** 3 good
**Contribution:** 3 good
**Rating:** 5
**Confidence:** 3

**Summary:**

This paper proposes a robustness-aware coreset selection (RCS) method, which is applied to accelerate adversarial contrast learning (ACL) in the absence of labeling information. Especially, the coreset searched by RCS minimizes the representation difference between the natural data and their adversarial examples, which is achieved by a greedy search method. And experimental results demonstrate that RCS can indeed speed up ACL without signiﬁcantly compromising the robustness transferability.

**Strengths:**

1. This paper is well-written and easy to follow.
2. The coreset searched by RCS is not only small in number but also beneficial in improving the adversarial robustness of representations.
3. Experimental results demonstrate that RCS can indeed speed up ACL without signiﬁcantly compromising the robustness transferability.


**Weaknesses:**

1. This paper points out that ACL with RCS trains the model on the previously selected coreset for $\lambda$ epochs, and for every $\lambda$ epochs a new coreset is selected. So, does the value of $\lambda$ affect the effectiveness as well as the efficiency of ACLs? It may be worthwhile for the authors to make further discussion.
2. In the experimental part, it can be found that the coreset searched by RCS without label information achieves better performance compared to the coreset searched by ACS, or even the whole dataset, as shown in Tables 12 and 13. How to explain this interesting phenomenon?
3. The core of this paper is the coreset found by RCS, so the authors should have given a more focused discussion, such as whether it is class-balanced and what the distribution of the subset is.


**Questions:**

Please refer to the weaknesses part.

**Limitations:**

Please refer to the weaknesses part.

---

> ### Author Rebuttal · Authors · 2023-08-04
>
> Many thanks for your positive and constructive comments! Please find our responses below.
>
> > 1. [Reply to W1] A larger $\lambda$ leads to more pre-training time and higher robust and standard test accuracy in downstream tasks.
>
> We pre-trained ResNet-18 on CIFAR-10 via ACL with RCS using $\lambda \in \\{ 10,20,50\\}$, $k=0.2$, and $\omega=100$. Then, we evaluate the performance on CIFAR-10 via SLF.
>
> | | | SLF on CIFAR-10 | SLF on CIFAR-10 |
> |---|---|---|---|
> | $\lambda$| Pre-training time (hours) | SA (\%) | RA (\%) |
> | 10 | 15.6 | 76.31 | 38.17 |
> | 20 | 13.0 | 75.96 | 37.21 |
> | 50 | 12.2 | 75.87 | 35.54 |
>
> The above table shows that a larger $\lambda$ leads to more pre-training time since the frequency of conducting coreset selection becomes larger.
>
> Meanwhile, a larger $\lambda$ can lead to higher robust and standard test accuracy in downstream tasks. This is because a larger $\lambda$ enables the coreset to be updated more timely to adapt to the latest state of the model and select data points that can help the model to improve its robustness.
>
> > 2. [Reply to W2] Thanks for pointing out this interesting phenomenon! We provide an explanation below.
>
> The phenomenon shown in Table 12-13 is that RCS can speed up Fast-AT and Free-AT while maintaining robust test accuracy. However, RCS sacrifices some natural test accuracy compared to the entire training set. For example, Fast-AT with RCS improves robust test accuracy by 0.02\% while degrading natural test accuracy by 2.73\% compared to Fast-AT on the entire set (Fast-AT-Entire).
>
> Here, we provide an explanation. We empirically find that the RD losses on the CIFAR-10 test set of ResNet-18 trained by Fast-AT-Entire, Fast-AT with ACS, and Fast-AT with RCS are 0.0394, 0.0423, and 0.0375, respectively. It indicates that our proposed RCS, whose objective is to find coresets that help minimize the RD loss, indeed minimizes the RD loss. Note that TRADES [1] proposed to obtain adversarial robustness by penalizing the KL divergence between natural data and its adversarial variant which equals our proposed RD loss.  Therefore, our proposed RCS can help maintain and even slightly improve adversarial robustness by selecting coresets that aim to minimize the RD loss.
>
> Besides, our proposed RCS actually utilizes the information of the entire training set during selecting a coreset. In particular, RCS dynamically updates the coreset according to the entire training set and the latest model parameters every $\lambda$ epoch. In this regard, RCS does not use less information compared to the entire training set. Therefore, RCS can achieve the robustness comparable to the entire training set.
>
> > 3. [Reply to W3] Thanks for your great suggestions! We demonstrate that RCS can select a coreset that is closer to the full training set than Random.
>
> Note that the imbalance ratio [2] is the ratio of the sample size of the largest majority class and that of the smallest minority class. Maximum mean discrepancy (MMD) [3] based on the Guassian kernel is a classical measurement of the distance between two distributions.
>
> The left panel of ***Figure G2*** in the "global" file shows that the corset selected by RCS is almost class-balanced since the imbalance ratio of RCS is slightly higher than 1.0. The right panel of ***Figure G2*** in the "global" file shows that RCS yields a lower MMD between the entire training set and the selected coreset compared to Random.
> Therefore, our quantitative analysis demonstrates that RCS generates a coreset that is closer to the entire training set than Random.
>
> *References*
>
> [1] Zhang, Hongyang, et al. "Theoretically principled trade-off between robustness and accuracy." International conference on machine learning. PMLR, 2019.\
> [2] Ortigosa-Hernández, J., Inza, I., & Lozano, J. A. (2017). Measuring the class-imbalance extent of multi-class problems. Pattern Recognition Letters, 98, 32-38.\
> [3] Gretton, A., Borgwardt, K. M., Rasch, M. J., Schölkopf, B., & Smola, A. (2012). A kernel two-sample test. The Journal of Machine Learning Research, 13(1), 723-773.

---

> > ### Comment · Reviewer_LfDi · 2023-08-21
> >
> > I thank the authors for their comprehensive responses and new results. After reading other reviewers' opinions, I decide to raise the score.

---

> > > ### Author Response · Authors · 2023-08-22
> > > **Thank you for your decision to raising the score**
> > >
> > > Dear Reviewer LfDi,
> > >
> > > Many thanks for acknowledging our responses & new results.
> > > We very appreciate your decision to raise the score.
> > >
> > > We noticed that your initial score is 5.
> > > Do you mean you want to increase your score more than 5?
> > >
> > > Thanks and best wishes,\
> > > Authors

---

> > > > ### Comment · Reviewer_LfDi · 2023-08-22
> > > >
> > > > Indeed, I chose to elevate the score to 6.

---

### Official Review · Reviewer_fJND · 2023-07-07

**Soundness:** 3 good
**Presentation:** 3 good
**Contribution:** 3 good
**Rating:** 6
**Confidence:** 4

**Summary:**

This paper proposes a coreset selection for efficient adversarial self-supervised learning. By selecting a coreset every epoch that can minimize the representation divergence for training, it maintains similar robustness performance despite a learning speed that is more than three times faster.

**Strengths:**

- It is technically sound. The fact that adversarial self-supervised learning (ASL) can be trained more than three times faster using coreset selection is especially remarkable.
- The proposed claims are well supported through theoretical analysis and experiments.

**Weaknesses:**

- The biggest weakness of the proposed method is that there isn't a significant difference compared to random selection. It utilizes more computation than random selection, but the gain in performance is small. It would be nice if the difference in computation compared to random selection could be explained.
- Moreover, the originality of the proposed method seems more like a simple application of applying the RD loss to previous work [1,2] rather than a new combination. If the author could explain this more clearly, I would be willing to revise my score.
- Furthermore, the one-step gradient approximation, warmup, last layer gradients, and adversarial example data approximation also seem to be techniques proposed in previous work, so there seems to be no originality in this regard.
- There seems to be a lack of justification that a coreset with a small representational divergence is sufficient to gain robustness. And there's not enough explanation on how the results can achieve comparable robustness.
- It would be helpful if there were explanations, perhaps utilizing representation visualization, as to what samples become the coreset selection, and why it helps with robustness.

[1] Kilamsetty et al., RETRIEVE: Coreset Selection for Efficient and Robust Semi-Supervised Learning

[2] Kilamsetty et al., Glister: Generalization based data subset selection for efficient and robust learning.


==After rebuttal
I changed my score to 6.

**Questions:**

None

**Limitations:**

The authors have well described their limitations.

---

> ### Author Rebuttal · Authors · 2023-08-04
>
> Many thanks for your positive and thoughtful comments! Please find our responses below.
>
> > 1. [Reply to W1] Our RCS obtains **substantial improvement** compared to random selection (Random).
>
> According to Figure 2, we highlight the performance gain of RCS in terms of robustness transferability from CIFAR-10 to STL10. Besides, in Figure 2-4, the solid line (results of RCS) is always far above the dotted line (results of Random), which validates that RCS consistently yields substantial improvement in standard and robust test accuracy compared to Random.
>
> In terms of computation, RCS spends slightly more pre-training time than Random since RCS needs to spend time in coreset selection every $\lambda$ epoch.
> |DynACL on CIFAR-10|||ALF on STL10|ALF on STL10|
> |---|---|---|---|---|
> |Subset fraction $k$|Method|Pre-training time (hours)|SA (\%)|RA (\%)|
> |0.05|Random|6.5|52.49|26.20|
> |0.05|RCS|7.6 (+1.1)|60.60 (**+8.11**)|32.35 (**+6.15**)|
> |0.1|Random|8.3|54.85|27.85|
> |0.1|RCS|9.4 (+1.1)|62.79 (**+8.21**)|33.89 (**+6.04**)|
> |0.2|Random|11.9|55.86|29.45|
> |0.2|RCS|13.1 (+1.2)|63.41 (**+7.55**)|34.76 (**+5.31**)|
> > 2. [Reply to W2] Simply applying our proposed RD loss to previous work [1,2] cannot obtain our proposed method.
>
> Simply applying RD loss to previous work [1,2] cannot obtain the objective function of RCS. To obtain a coreset, [1,2] formulates a bi-level optimization where they first obtain the parameters via minimizing the inner loss and then find the coreset via minimizing the outer loss. If we simply replace the outer loss of [1,2] with RD loss, the objective function still does not apply to ACL since the inner loss of [1,2] is a natural loss and requires labels. Therefore, to adapt the CS to speed up ACL, we also utilize the ACL loss as inner loss.
>
> Besides, we cannot simply apply the RD loss to the greedy search algorithm in [1,2] for solving our proposed RCS. Since the objective function of RCS is different from [1,2], we are not sure whether the greedy search is applicable to our proposed RCS with an optimality guarantee. Therefore, we provide Theorem 1 and 2, which are basically based on the mathematical knowledge of monotonicity and $\gamma$-weakly submodularity, to theoretically prove that the algorithm of greedy search can be applied to our proposed RCS with an optimality guarantee. Finally, we proposed our algorithm of RCS via greedy search.
> > 3. [Reply to W3] Besides these four tricks, we propose an unique trick to enable efficient RCS on large-scale datasets with limited GPU memory in Appendix B.1.
>
> RCS on large-scale datasets such as ImageNet-1K needs a large amount of GPU memory to calculate and save the gradient for each minibatch of training data on GPU as the first step (Line 5–7 in Algorithm 1). When the GPU memory is limited, there exists an issue that we are unable to save all the gradients on GPU.
>
> To solve this issue, we split the entire training set into several training subsets. Then, we conduct RCS on each training subset and combines the coresets from each training subset together as the final coresets for robust pre-training. In this way, we enable RCS to efficiently search for the coresets from large-scale datasets with limited GPU memory. We apply this trick to the experiments regarding ACL on ImageNet-1K in Section 4.2 and SAT on ImageNet-1K in Section 4.3. We believe this trick can help the implementation of RCS on large-scale datasets in a computational resource-restricted environment.
> > 4. [Reply to W4] We show that a lower RD loss (i.e., the KL divergence between natural data and its adversarial variant) on the test set corresponds to better adversarial robustness in Figure 5 (in Appendix).
>
> Here, we copy the results of Figure 5 to help explain.
> |ACL on CIFAR-10||SLF on CIFAR-10|SLF on CIFAR-10|
> |---|---|---|---|
> | | RD loss (lower is better)|SA (\%)|RA (\%)|
> |Entire|0.1243|78.87|39.19|
> |Random-0.05|0.3357|67.45|22.96|
> |RCS-0.05|0.1730|72.56|32.49|
> |Random-0.1|0.3094|70.68|27.19|
> |RCS-0.1|0.1695|74.67|34.30|
> |Random-0.2|0.2333|72.01|29.87|
> |RCS-0.2|0.1664|75.96|37.21|
>
> The table shows that our proposed RCS, whose objective is to find coresets that help minimize the RD loss, indeed minimizes the RD loss. Meanwhile, it empirically shows that a lower RD loss corresponds to higher robust test accuracy. Note that TRADES [1] proposed to obtain adversarial robustness by penalizing the KL divergence between natural data and its adversarial variant which equals our proposed RD loss. Therefore, our proposed RCS can help maintain adversarial robustness by selecting coresets that aim to minimize the RD loss.
> > 5. [Reply to W5] Thanks for your suggestions! We provide a visualisation analysis as follows.
>
> We count the frequency of each training sample in the CIFAR-10 dataset being selected into the coreset. Then, we visualize the top-5 most-frequently selected (MFS) data and the top-5 least-frequently selected (LFS) data in ***Figure G1*** in the "global" file.
>
> ***Figure G1*** shows that, compared to LFS data, MFS data are images whose backgrounds are more complex and are more difficult to be distinguished from the subject. Recent work [3,4] has shown that exempting the representations from the nuisanse style factors such as the background factor can improve robustness against distribution shifts. RCS prefers to select the images of complex backgrounds helps the model learn representations that are independent of the background factors, thus helping maintain robustness against adversarial pertubations.
>
> References\
> [1] Kilamsetty et al., RETRIEVE: Coreset Selection for Efficient and Robust Semi-Supervised Learning\
> [2] Kilamsetty et al., Glister: Generalization based data subset selection for efficient and robust learning\
> [3] Representation learning via invariant causal mechanisms. Mitrovic et al., ICLR 2021\
> [4] Invariant risk minimization. Arjovsky et al., 2020\
> [5] Theoretically principled trade-off between robustness and accuracy. Zhang et al., ICML 2019

---

> > ### Comment · Reviewer_fJND · 2023-08-14
> >
> > Thank you to the authors for their comprehensive response and additional experiments. Most of my concerns have been addressed, so I have decided to increase my score to 6.

---

### Author Rebuttal · Authors · 2023-08-04

[**Rebuttal Highlights**]

Many thanks for all reviewers' supportive and constructive comments!

Following reviewers' suggestions, we uploaded extensive ***Figure G1*** and ***Figure G2*** in the "**global**" file to provide a focused discussion of the coreset.

> 1. [For Reviewer **fJND**] In ***Figure G1***, we provide a visualization analysis to interpret why the coreset selected by RCS can help maintain robustness.

We count the frequency of each training sample in the CIFAR-10 dataset being selected into the coreset. Then, we visualize the top-5 most-frequently selected (MFS) data and the top-5 least-frequently selected (LFS) data in ***Figure G1***.

***Figure G1*** shows that, compared to LFS data, MFS data are images whose backgrounds are more complex and are more difficult to be differentiated from the subject. Recent work [1,2] has shown that exempting the representations from the nuisanse style factors such as the background factor can improve the robustness against distribution shifts. RCS prefers to select the images of complex backgrounds helps the model learn representations that are independent of the background factors, thus helping maintain robustness against adversarial pertubations.

> 2. [For Reviewer **LfDi**] In ***Figure G2***, we demonstrate that RCS can select a coreset that is closer to the full training set than Random.

Note that the imbalance ratio [3] is the ratio of the sample size of the largest majority class and that of the smallest minority class. Maximum mean discrepancy (MMD) [4] based on the Guassian kernel is a classical measurement of the distance between two distributions.

The left panel of ***Figure G2*** shows that the corset selected by RCS is almost class-balanced since the imbalance ratio of RCS is slightly higher than 1.0. The right panel of ***Figure G2*** shows that RCS yields a lower MMD between the entire training set and the selected coreset compared to Random. Therefore, our quantitative analysis demonstrates that RCS selects a coreset that is closer to the entire training set than Random.

*References*

[1] Representation learning via invariant causal mechanisms. Mitrovic et al., ICLR 2021.\
[2] Invariant risk minimization. Arjovsky et al., 2020.\
[3] Ortigosa-Hernández, J., Inza, I., & Lozano, J. A. (2017). Measuring the class-imbalance extent of multi-class problems. Pattern Recognition Letters, 98, 32-38.\
[4] Gretton, A., Borgwardt, K. M., Rasch, M. J., Schölkopf, B., & Smola, A. (2012). A kernel two-sample test. The Journal of Machine Learning Research, 13(1), 723-773.

---

### Decision · Program_Chairs · 2023-09-21

**Decision:**

Accept (spotlight)

**Comment:**

There is consensus among all four expert referees that the paper should be accepted. During the initial reviewing phase, there were some questions raised, mainly about the experiments. The rebuttal provided further clarifications, which should be incorporated in the final version of the paper.